# Complexity of Highly Parallel
# Non-Smooth Convex Optimization

**Sébastien Bubeck**
Microsoft Research
sebubeck@microsoft.com

**Qijia Jiang**
Stanford University
qjiang2@stanford.edu

**Yin Tat Lee**
University of Washington
& Microsoft Research
yintat@uw.edu

**Yuanzhi Li**
Stanford University
yuanzhil@stanford.edu

**Aaron Sidford**
Stanford University
sidford@stanford.edu

## Abstract

A landmark result of non-smooth convex optimization is that gradient descent is an optimal algorithm whenever the number of computed gradients is smaller than the dimension $d$. In this paper we study the extension of this result to the parallel optimization setting. Namely we consider optimization algorithms interacting with a highly parallel gradient oracle, that is one that can answer $\mathrm{poly}(d)$ gradient queries in parallel. We show that in this case gradient descent is optimal only up to $\widetilde{O}(\sqrt{d})$ rounds of interactions with the oracle. The lower bound improves upon a decades old construction by Nemirovski which proves optimality only up to $d^{1/3}$ rounds (as recently observed by Balkanski and Singer), and the suboptimality of gradient descent after $\sqrt{d}$ rounds was already observed by Duchi, Bartlett and Wainwright. In the latter regime we propose a new method with improved complexity, which we conjecture to be optimal. The analysis of this new method is based upon a generalized version of the recent results on optimal acceleration for highly smooth convex optimization.

## 1 Introduction

Much of the research in convex optimization has focused on the *oracle model*, where an algorithm optimizing some objective function $f : \mathbb{R}^d \to \mathbb{R}$ does so by sequential interaction with, e.g., a gradient oracle (given a query $x \in \mathbb{R}^d$, the oracle returns $\nabla f(x)$), [Nemirovski and Yudin, 1983, Nesterov, 2004, Bubeck, 2015].[1] In the early 1990s, Arkadi Nemirovski introduced the parallel version of this problem [Nemirovski, 1994]: instead of submitting queries one by one sequentially, the algorithm can submit in parallel up to $Q \geq 1$ queries. We refer to the *depth* of such a parallel algorithm as the number of rounds of interaction with the oracle, and the *work* as the total number of queries (in particular work $\leq Q \times$ depth). In this paper we study the optimal depth achievable for highly parallel algorithms, namely we consider the regime $Q = \mathrm{poly}(d)$. We focus on non-smooth convex optimization, that is we want to optimize a Lipschitz, convex function $f$ on the unit Euclidean ball.

Our key result is a new form a quadratic acceleration: while for purely sequential methods the critical depth at which one can improve upon local search is $\widetilde{O}(d)$, we show that in the highly parallel regime the critical depth is $\widetilde{O}(\sqrt{d})$.

## 1.1 Classical optimality results

Classically, when $Q = 1$, it is known that gradient descent's query complexity is *order optimal* for any target accuracy $\varepsilon$ in the range $\left[d^{-1/2}, 1\right]$. More precisely, it is known that the query complexity of gradient descent is $O(1/\varepsilon^2)$ and that for any $\varepsilon$ in the range $\left[d^{-1/2}, 1\right]$, and for any algorithm, there exists a Lipschitz and convex function $f$ on which the number of oracle queries the algorithm makes to achieve additive $\varepsilon$ accuracy is $\Omega(1/\varepsilon^2)$. Furthermore, whenever $\varepsilon$ is smaller than $d^{-1/2}$ there exists a better algorithm (i.e., with smaller depth), namely the center of gravity whose depth is $O(d \log(1/\varepsilon))$. Consequently, an alternative formulation of these results is that, for $Q = 1$, gradient descent is order optimal if and only if the depth is smaller than $\widetilde{O}(d)$. (See previously cited references for the exact statements.)

## 1.2 Optimality for highly parallel algorithms

The main result of this paper is to show that in the highly parallel regime ($Q = \mathrm{poly}(d)$), gradient descent is order optimal if and only if the depth is smaller than $\widetilde{O}(\sqrt{d})$. Thus one has a "quadratic" improvement over the purely sequential setting in terms of the critical depth at which naive local search becomes suboptimal.

The *only if* part of the above statement follows from Duchi et al. [2012], where *randomized smoothing* with accelerated gradient descent was proposed (henceforth referred to as distributed randomized smoothing [Scaman et al., 2018]), and shown to achieve depth $d^{1/4}/\varepsilon$, which is order better than $1/\varepsilon^2$ exactly when the latter is equal to $\sqrt{d}$. A first key contribution of our work is a matching lower bound showing that, when the depth is smaller than $\widetilde{O}(\sqrt{d})$, no significant improvement over gradient descent is possible, i.e. $Q = 1$ and $Q = \mathrm{poly}(d)$ have essentially the same power. Importantly we note that our lower bound applies to randomized algorithms. The previous state of the art lower bound was that gradient descent is optimal up to depth $\widetilde{O}(d^{1/3})$ [Balkanski and Singer, 2018]. In fact the construction in the latter paper is exactly the same as the original construction of Nemirovski in [Nemirovski, 1994] (however the final statements are different, as Nemirovski was concerned with an $\ell_\infty$ setting instead of $\ell_2$, see also Diakonikolas and Guzmán [2018] for more results about non-Euclidean setups).

A second key contribution of this work is to improve the state of the art complexity of parallel algorithms with depth between $\sqrt{d}$ and $d$. Improving the depth $d^{1/4}/\varepsilon$ of Duchi et al. [2012] was explicitly mentioned as an open problem by Scaman et al. [2018]. Leveraging the recent higher order acceleration schemes of Gasnikov et al. [2018], Jiang et al. [2018], Bubeck et al. [2018], we propose a new method with depth $d^{1/3}/\varepsilon^{2/3}$. This means that for any value of $\varepsilon$ in the range $\left[d^{-1}, d^{-1/4}\right]$ there is an algorithm that is order better than *both* gradient descent and center of gravity. Moreover we conjecture that the depth $d^{1/3}/\varepsilon^{2/3}$ is in fact optimal for any $\varepsilon$ in this range (as the arguments of Section 2.3 would imply this if a similar argument could be made for $\delta = \varepsilon$, i.e. a smaller walling radius). We leave this question, the optimality of the center of gravity method for small $\varepsilon < 1/\mathrm{poly}(d)$, and the optimal work among optimal depth algorithms, for future works.

## 1.3 Related works

Though Nemirovski's prescient work stood alone for decades, more recently the subfield of parallel/distributed optimization is booming, propelled by problems in machine learning, see e.g., [Boyd et al., 2011]. Chief among those problems is how to leverage mini-batches in stochastic gradient descent as efficiently as possible [Dekel et al., 2012]. The literature on this topic is sprawling, see for example [Duchi et al., 2018] which studies the total work achievable in parallel stochastic convex optimization, or [Zhang and Xiao, 2018] where the stochastic assumptions are leveraged to take advantage of second order information. More directly related to our work is [Nemirovski, 1994, Diakonikolas and Guzmán, 2018, Balkanski and Singer, 2018] from the lower bound side (we directly

improve upon the result in the latter paper), and [Duchi et al., 2012, Scaman et al., 2018] from the upper bound side (we directly improve upon the depth provided by the algorithms in those works).

## 2 Lower bound

Fix $\varepsilon > 0$ such that $1/\varepsilon^2 = \widetilde{O}(\sqrt{d})$. In this section we construct a random function $f$ such that, for any deterministic algorithm with depth $O(1/\varepsilon^2)$ and total work $\mathrm{poly}(d)$, the output point $x$ is such that $\mathbb{E}[f(x) - f^*] > \varepsilon$, where the expectation is with respect to the random function $f$, and $f^*$ denotes the minimum value of $f$ on the unit centered Euclidean ball. Note that by the minimax theorem, this implies that for any randomized algorithm there exists a deterministic function such that the same conclusion applies. Formally, we prove the following:

**Theorem 1 (Lower Bound)** *Let $\rho \in (0, 1)$ and $C = 12 + 4\log_d(Q/\rho)$. Further, assume that it holds that $\log(N)N\sqrt{C\log(d)/d} \leq \frac{1}{4}$ (i.e., $N \lesssim \sqrt{d/\log^3(d)}$). Fix a randomized algorithm that queries at most $Q$ points per iteration (both function value and gradient), and that runs for at most $N$ iterations. Then, with probability at least $1 - \rho$, when run on the shielded Nemirovski function $f$ (see Section 2.3 and Section 2.4)) one has for any queried point: $f(x) - f^* \geq \frac{1}{4\sqrt{N}}$.*

The details of the proof of this theorem are deferred to Appendix A. In the remainder of this section we instead provide a sketch of its proof. We first recall in Section 2.1 why, for purely sequential algorithms, the above statement holds true, and in fact one can even replace $\sqrt{d}$ by $d$ in this case (this construction goes back to [Yudin and Nemirovski, 1976], see also [Nemirovski and Yudin, 1983]). Next, in Section 2.2 we explain Nemirovski [1994]'s construction, which yields a weaker version of the above statement, with $\sqrt{d}$ replaced by $d^{1/3}$ (as rediscovered by [Balkanski and Singer, 2018]). We then explain in Section 2.3 our key new construction, a type of *shielding* operation. Finally, we conclude the proof sketch in Section 2.4.

For the rest of the section we let $v_1, \ldots, v_N$ denote $N$ random orthonormal vectors in $\mathbb{R}^d$ (in particular $N \leq d$), and $x^* = -\frac{1}{\sqrt{N}} \sum_{i=1}^{N} v_i$. We define the *Nemirovski function* with parameter $\gamma \geq 0$ by:

$$\mathcal{N}(x) = \max_{i \in [N]} \left\{ v_i \cdot x - i\gamma \right\},$$

Note that

$$\mathcal{N}^* \leq \mathcal{N}(x^*) \leq -\frac{1}{\sqrt{N}}. \tag{1}$$

### 2.1 The classical argument

We consider here the Nemirovski function with parameter $\gamma = 0$. Each gradient query reveals a single vector in the collection of the $v_i$, so after $N/2$ iterations one might know say $v_1, \ldots, v_{N/2}$, but the rest remain unknown (or in other words they remain random orthonormal vectors in $\mathrm{span}(v_1, \ldots, v_{N/2})^\perp$). Thus for any output $x$ that depends on only $N/2$ queries, one has $\mathbb{E}[\mathcal{N}(x)] \geq 0$ (formally this inequality follows from Jensen's inequality and the tower rule). Thus, together with (1), it follows that $\mathbb{E}[\mathcal{N}(x) - \mathcal{N}^*] \geq 1/\sqrt{N}$. In other words the best rate of convergence of sequential methods is $1/\sqrt{N}$, provided that $N \leq d$.

### 2.2 The basic parallel argument

We consider here the Nemirovski function with parameter $\gamma = C\sqrt{\log(d)/d}$ for some large enough constant $C$ (more precisely that constant $C$ depends on the exponent in the $\mathrm{poly}(d)$ number of allowed queries per round). The key observation is as follows: Imagine that the algorithm has already discovered $v_1, \ldots, v_{i-1}$. Then for any set of $\mathrm{poly}(d)$ queries, with high probability with respect to the random draw of $v_i, \ldots, v_N$, one has that the inner product of any of those vectors with any of the queried points is in $[-\gamma/2, \gamma/2]$ (using both basic concentration of measure on the sphere, and a union bound). Thus the maximum $\gamma$ in the definition of $\mathcal{N}$ is attained at some index $\leq i$. This means that this set of $\mathrm{poly}(d)$ queries can only reveal $v_i$, and not any of the $v_j, j > i$. Thus after $N - 1$ rounds we know that with high probability any output $x$ satisfies $\mathcal{N}(x) \geq v_N \cdot x - N\gamma \geq -(N+1)\gamma$ (since

$v_N$ is a random direction orthogonal to $\text{span}(v_1, \ldots, v_{N-1})$ and $x$ only depends on $v_1, \ldots, v_{N-1}$. Thus we obtain that the suboptimality gap is $\frac{1}{\sqrt{N}} - (N+1)\gamma$. Let us assume that

$$N^{3/2} \leq \frac{1}{2\gamma} \,, \tag{2}$$

i.e., $N = \widetilde{O}(d^{1/3})$ (since $\gamma = C\sqrt{\log(d)/d}$). Then one has that the best rate of convergence with a highly parallel algorithm is $\Omega(1/\sqrt{N})$ (i.e., the same as with purely sequential methods).

## 2.3 The wall function

Our new idea to improve upon Nemirovski's construction is to introduce a new random *wall function* $\mathcal{W}$ (with parameter $\delta > 0$), where the randomness come from $v_1, \ldots, v_N$. Our new random hard function, which we term *shielded-Nemirovski function*, is then defined by:

$$f(x) = \max\{\mathcal{N}(x), \mathcal{W}(x)\} \,.$$

We construct the convex function $\mathcal{W}$ so that one can essentially repeat the argument of Section 2.2 with a smaller value of $\gamma$ (the parameter in the Nemirovski function), so that the condition (2) becomes less restrictive and allows to take $N$ as large as $\widetilde{O}(\sqrt{d})$.

Roughly speaking the wall function will satisfy the following properties:

1. The value of $\mathcal{W}$ at $x^*$ is small, namely $\mathcal{W}(x^*) \leq -\frac{1}{\sqrt{N}}$.

2. The value of $\mathcal{W}$ at "most" vectors $x$ with $\|x\| \geq \delta$ is large, namely $\mathcal{W}(x) \geq \mathcal{N}(x)$, and moreover it is does not depend on the collection $v_i$ (in fact at most points we will have the simple formula $\mathcal{W}(x) = 2\|x\|^{1+\alpha}$, for some small $\alpha$ that depends on $\delta$, to be defined later).

The key argument is that, by property 2, one can expect (roughly) that information on the random collection of $v_i's$ can only be obtained by querying points of norm smaller than $\delta$. This means that one can repeat the argument of Section 2.2 with a smaller value of $\gamma$, namely $\gamma = \delta \cdot C\sqrt{\log(d)/d}$. In turn the condition (2) now becomes $N = \widetilde{O}\left(d^{1/3}/\delta^{2/3}\right)$. Due to convexity of $\mathcal{W}$, there is a tension between property 1 and 2, so that one cannot take $\delta$ too small. We will show below that it is possible to take $\delta = \sqrt{N/d}$. In turn this means that the argument proves that $1/\sqrt{N}$ is the best possible rate, up to $N = \widetilde{O}(\sqrt{d})$.

The above argument is imprecise because the meaning of "most" in property 2 is unclear. A more precise formulation of the required property is as follows:

2' Let $x = w + z$ with $w \in V_i$ and $z \in V_i^\perp$ where $V_i = \text{span}(v_1, \ldots, v_i)$. Assume that $\|z\| \geq \delta$, then the total variation distance between the conditional distribution of $v_{i+1}, \ldots, v_N$ given $\nabla\mathcal{W}(x)$ (and $\mathcal{W}(x)$) and the unconditional distribution is polynomially small in $d$ with high probability (here high probability is with respect to the realization of $\nabla\mathcal{W}(x)$ and $\mathcal{W}(x)$, see below for an additional comment about such conditional reasoning). Moreover if the argmax in the definition of $\mathcal{N}(x)$ is attained at some index $> i$, then $\mathcal{W}(x) \geq \mathcal{N}(x)$.

Given both property 1 and 2' it is actually easy to formalize the whole argument. We do so by consdering a game between Alice, who is choosing the query points, and Bob who is choosing the random vectors $v_1, \ldots, v_N$. Moreover, to clarify the reasoning about conditional distributions, Bob will resample the vectors $v_i, \ldots, v_N$ at the beginning of the $i^{th}$ round of interaction, so that one explicitly does not have any information about those vectors given the first $i-1$ rounds of interaction. Then we argue that with high probability all the oracle answers remain consistent throughout this resampling process. See Appendix A for the details. Next we explain how to build $\mathcal{W}$ so as to satisfy property 1 and 2'.

## 2.4  Building the wall

Let $h(x) = 2\|x\|^{1+\alpha}$ be the basic building block of the wall. Consider the correlation cones:

$$C_i = \left\{ x \in \mathbb{R}^d : \left| v_i \cdot \frac{x}{\|x\|} \right| \geq C\sqrt{\frac{\log(d)}{d}} \right\} .$$

Note that for any fixed query $x$, the probability (with respect to the random draw of $v_i$) that $x$ is in $C_i$ is polynomially small in $d$. We now define the wall $\mathcal{W}$ as follows: it is equal to the function $h$ outside of the correlation cones and the ball of radius $\delta$, and it is extended by convexity to the rest of the unit ball. In other words, let $\Omega = \{ x \in \mathbb{R}^d : \|x\| \in [\delta, 1] \text{ and } x \notin C_i \text{ for all } i \in [N] \}$, and

$$\mathcal{W}(x) = \max_{y \in \Omega} \{ h(y) + \nabla h(y) \cdot (x - y) \} .$$

Let us first prove property 1:

**Lemma 2** *Let* $\alpha = \frac{1}{\log_2(1/\delta)} \leq 1$, *and* $\frac{\delta}{\log_2(1/\delta)} = 4C\sqrt{\frac{N \log(d)}{d}} + \frac{1}{\sqrt{N}}$. *Then* $\mathcal{W}(x^*) \leq -\frac{1}{\sqrt{N}}$.

**Proof**  One has $\nabla h(y) = 2(1+\alpha)\frac{y}{\|y\|^{1-\alpha}}$ and thus

$$h(y) + \nabla h(y) \cdot (x - y) = -2\alpha\|y\|^{1+\alpha} + 2(1+\alpha)\frac{y \cdot x}{\|y\|^{1-\alpha}} . \tag{3}$$

Moreover for any $y \in \Omega$ one has:

$$|y \cdot x^*| \leq \frac{1}{\sqrt{N}} \sum_{i=1}^{N} |y \cdot v_i| \leq C\sqrt{\frac{N \log(d)}{d}} \cdot \|y\| .$$

Thus for any $y \in \Omega$ we have:

$$h(y) + \nabla h(y) \cdot (x^* - y) \leq -2\alpha\delta^{1+\alpha} + 2(1+\alpha)C\sqrt{\frac{N \log(d)}{d}} .$$

The proof is straightforwardly concluded by using the values of $\alpha$ and $\delta$. ∎

Next we prove a simple formula for $\mathcal{W}(x)$ in the context of property $2'$. More precisely we assume that $x = w + z$ with $w \in V_i$ and $z \in V_i^\perp$ with $z \notin C_j$ for any $j > i$. Note that for any fixed $z$, the latter condition happens with high probability with respect to the random draw of $v_{i+1}, \ldots, v_N$.

**Lemma 3** *Let* $x = w + z$ *with* $w \in V_i$ *and* $z \in V_i^\perp$ *with* $z \notin C_j$ *for any* $j > i$. *Then one has:*

$$\mathcal{W}(x) = \max_{a,b \in \mathbb{R}_+ : a^2 + b^2 \in [\delta^2, 1]} \left\{ -2\alpha(a^2 + b^2)^{\frac{1+\alpha}{2}} + 2\frac{1+\alpha}{(a^2+b^2)^{\frac{1-\alpha}{2}}} \left( \max_{y \in \widetilde{\Omega}, \|y\|=a} y \cdot w + b\|z\| \right) \right\} ,$$

*where* $\widetilde{\Omega} = \{ x \in V_i : x \notin C_j \text{ for all } j \in [i] \}$ *and we use the convention that the maximum of an empty set is* $-\infty$.

**Proof**  Recall (3), and let us optimize over $y \in \Omega$ subject to $\|P_{V_i} y\| = a$ and $\|P_{V_i^\perp} y\| = b$ for some $a, b$ such that $a^2 + b^2 \in [\delta^2, 1]$. Note that in fact there is an upper bound constraint on $a$ for such a $y$ to exists (for if the projection of $y$ onto $V_i$ is large, then necessarily $y$ must be in one of the correlation cones), which we can ignore thanks to the convention choice for the maximum of an empty set. Thus the only calculation we have to do is to verify that:

$$\max_{y \in \Omega : \|P_{V_i} y\|=a \text{ and } \|P_{V_i^\perp} y\|=b} y \cdot x = \max_{y \in \widetilde{\Omega}, \|y\|=a} y \cdot w + b\|z\| .$$

Note that $y \cdot x = P_{V_i} y \cdot w + P_{V_i^\perp} y \cdot z$. Thus the right hand side is clearly an upper bound on the left hand side (note that $P_{V_i} y \in \widetilde{\Omega}$). To see that it is also a lower bound take $y = y' + b\frac{z}{\|z\|}$ for some arbitrary $y' \in \widetilde{\Omega}$ with $\|y'\| = a$, and note that $y \in \Omega$ (in particular using the assumption on $z$) with

$\|P_{V_i} y\| = a$ and $\|P_{V_i^\perp} y\| = b$. ∎

The key point of the formula given in Lemma 3 is that it does not depend on $v_{i+1}, \ldots, v_N$. Thus when the algorithm queries the point $x$ and obtains the above value for $\mathcal{W}(x)$ (and the corresponding gradient), the only information that it obtains is that $z \notin C_j$ for any $j > i$. Since the latter condition holds with high probability, the algorithm essentially learns nothing (more precisely the conditional distribution of $v_{i+1}, \ldots, v_N$ only changes by $1/\mathrm{poly}(d)$ compared to the unconditional distribution).

Thus to complete the proof of property $2'$ it only remains to show that if $\|z\| \geq \delta$ and the argmax in $\mathcal{N}(x)$ is attained at an index $> i$, then the formula in Lemma 3 is larger than $\mathcal{N}(x)$. By taking $a = 0$ and $b = \delta$ one obtains that this formula is equal to (using also the values assigned to $\alpha$ in Lemma 2):

$$-2\alpha\delta^{1+\alpha} + 2\frac{1+\alpha}{\delta^{1-\alpha}}\delta\|z\| = -\alpha\delta + (1+\alpha)\|z\| \geq \|z\|.$$

On the other hand one has (by assumption that the $\arg\max$ index is $> i$)

$$\mathcal{N}(x) = \max_{j>i}\{v_j \cdot x - j\gamma\} \leq \|z\|.$$

This concludes the proof of property $2'$, and in turn concludes the proof sketch of our lower bound.

## 3  Upper bound

Here we present our highly parallel optimization procedure. Throughout this section we let $f : \mathbb{R}^d \to \mathbb{R}$ denote a differentiable $L$-Lipschitz function that obtains its minimum value at $x^* \in \mathbb{R}^d$ with $\|x^*\|_2 \leq R$. The main result of this section is the following theorem, which provides an $\widetilde{O}(d^{1/3}/\varepsilon^{2/3})$-depth highly-parallel algorithm that computes an $\varepsilon$-optimal point with high probability.

**Theorem 4 (Highly Parallel Function Minimization)** *There is a randomized highly-parallel algorithm which given any differentiable L-Lipschitz $f : \mathbb{R}^d \to \mathbb{R}$ minimized at $x^*$ with $\|x^*\| \leq R$ computes with probability $1 - \nu$ a point $x \in \mathbb{R}^d$ with $f(x) - f(x^*) \leq \varepsilon$ in depth $\widetilde{O}(d^{1/3}(LR/\varepsilon)^{2/3})$ and work $\widetilde{O}(d^{4/3}(LR/\varepsilon)^{8/3})$ where $\widetilde{O}(\cdot)$ hides factors polylogarithmic in $d,\varepsilon,L,R$, and $\nu^{-1}$.*

Our starting point for obtaining this result are the $O(d^{1/4}/\varepsilon)$ depth highly parallel algorithms of [Duchi et al., 2012]. This paper considers the convolution of $f$ with simple functions, e.g. Gaussians and uniform distributions, and shows this preserves the convexity and continuity of $f$ while improving the smoothness and thereby enables methods like accelerated gradient descent (AGD) to run efficiently. Since the convolved function can be accessed efficiently in parallel by random sampling, working with the convolved function is comparable to working with the original function in terms of query depth (up to the sampling error). Consequently, the paper achieves its depth bound by trading off the error induced by convolution with the depth improvements gained from stochastic variants of AGD.

To improve upon this bound, we apply a similar approach of working with the convolution of $f$ with a Gaussian. However, instead of applying standard stochastic AGD we consider accelerated methods which build a more sophisticated model of the convolved function in parallel. Instead of using random sampling to approximate only the gradient of the convolved function, we obtain our improvements by using random sampling to glean more local information with each highly-parallel query and then use this to minimize the convolved function at an accelerated rate.

To enable the use of these more sophisticated models we develop a general acceleration framework that allows us to leverage any subroutine for approximate minimization a local model/approximate gradient computations into an accelerated minimization scheme. We believe this framework is of independent interest, as we show that we can analyze the performance of this method just in terms of simple quantities regarding the local model. This framework is discussed in Section 3.1 and in Appendix C where we show how it generalizes multiple previous results on near-optimal acceleration.

Using this framework, proving Theorem 4 reduces to showing that we can minimize high quality local models of the convolved function. Interestingly, it is possible to nearly obtain this result by simply random sampling to estimate all derivatives up to some order $k$ and then use this to minimize a regularized $k$-th order Taylor approximation to the function. Near-optimal convergence for such methods under Lipschitz bounds on the $k$-th derivatives were recently given by [Gasnikov et al., 2018, Jiang et al., 2018, Bubeck et al., 2018] (and follow from our framework). This approach can be shown

to give a highly-parallel algorithm of depth $\widetilde{O}(d^{1/3+c}/\varepsilon^{2/3})$ for any $c > 0$ (with an appropriately large $k$). Unfortunately, the work of these methods is $O(d^{\mathrm{poly}(1/c)})$ and expensive for small $c$.

To overcome this limitation, we leverage the full power of our acceleration framework and instead show that we can randomly sample to build a model of the convolved function accurate within a ball of sufficiently large radius. In Section 3.2 we bound this quality of approximation and show that this local model can be be optimized to sufficient accuracy efficiently. By combining this result with our framework we prove Theorem 4. We believe this demonstrates the utility of our general acceleration scheme and we plan to further explore its implications in future work.

## 3.1 Acceleration framework

Here we provide a general framework for accelerated convex function minimization. Throughout this section we assume that there is a twice-differentiable convex function $g : \mathbb{R}^d \to \mathbb{R}$ given by an *approximate proximal step oracle* and an *approximate gradient oracle* defined as follows.

**Definition 5 (Approximate Proximal Step Oracle)** *Let $\omega : \mathbb{R}_+ \to \mathbb{R}_+$ be a non-decreasing function, $\delta \geq 0$, and $\alpha \in [0,1)$. We call $\mathcal{T}_{\mathrm{prox}}$ an $(\alpha, \delta)$-approximate $\omega$-proximal step oracle for $g : \mathbb{R}^d \to \mathbb{R}$ if, for all $x \in \mathbb{R}^d$, when queried at $x \in \mathbb{R}^d$ the oracle returns $y = \mathcal{T}_{\mathrm{prox}}(x)$ such that*

$$\|\nabla g(y) + \omega(\|y - x\|)(y - x)\| \leq \alpha \cdot \omega(\|y - x\|)\|y - x\| + \delta . \tag{4}$$

**Definition 6 (Approximate Gradient Oracle)** *We call $\mathcal{T}_{\mathrm{grad}}$ an $\delta$-approximate gradient oracle for $g : \mathbb{R}^d \to \mathbb{R}$ if when queried at $x \in \mathbb{R}^d$ the oracle returns $v = \mathcal{T}_{\mathrm{grad}}(x)$ such that $\|v - \nabla g(x)\| \leq \delta$.*

We show that there is an efficient accelerated optimization algorithm for minimizing $g$ using only these oracles. Its performance is encapsulated by the following theorem.

**Theorem 7 (Acceleration Framework)** *Let $g : \mathbb{R}^d \to \mathbb{R}$ be a convex twice-differentiable function minimized at $x^*$ with $\|x^*\| \leq R$, $\varepsilon > 0$, $\alpha \in [0,1)$, and $\gamma \geq 1$ such that $128\alpha\gamma^2 \leq 1$. Further, let $\omega : \mathbb{R}_+ \to \mathbb{R}_+$ be a monotonically increasing continuously differentiable function with $0 < \omega'(s) \leq \gamma \cdot \omega(s)/s$ for all $s > 0$. There is an algorithm which for all $k$ computes a point $y_k$ with*

$$g(y_k) - g^* \leq \max\left\{ \varepsilon , \ \frac{32 \cdot \omega\left(\frac{40\|x^*\|}{k^{3/2}}\right)\|x^*\|^2}{k^2} \right\}$$

*using $k(6 + \log_2[10^{20}\gamma^6 R^2 \cdot \omega(10^5\gamma^2 R) \cdot \varepsilon^{-1}])^2$ queries to a $(\alpha, \delta)$-approximate $\omega$-proximal step oracle for $g$ and a $\delta$-approximate gradient oracle for $g$ provided that both $\delta \leq \varepsilon/[10^{20}\gamma^2 R]$ and $\varepsilon \leq 10^{20}\gamma^4 R^3 \omega(80R)$.*

This theorem generalizes multiple accelerated methods (up to polylogarithmic factors) and sheds light on the rates of these methods (See Appendix C for applications). For example, choosing $\omega(x) \stackrel{\mathrm{def}}{=} \frac{L}{2}$ and $\mathcal{T}_{\mathrm{prox}}(x) = x - \frac{1}{L}\nabla f(x)$ recovers standard accelerated minimization of $L$-smooth functions, choosing $\omega(x) \stackrel{\mathrm{def}}{=} \frac{L}{2}$ and $\mathcal{T}_{\mathrm{prox}}(x) \approx \arg\min_y g(y) + \frac{L}{2}\|y - x\|^2$ recovers a variant of approximate proximal point [Frostig et al.] and Catalyst [Lin et al., 2015], and choosing $\omega(x) \stackrel{\mathrm{def}}{=} \frac{L_p \cdot (p+1)}{p!}x^{p-1}$ and $\mathcal{T}_{\mathrm{prox}}(x) = \arg\min_y \ g_p(y;x) + \frac{L_p}{p!}\|y - x\|^{p+1}$ where $g_p(y;x)$ is the value of the $p$'th order Taylor approximation of $g$ about $x$ evaluated at $y$ recovers highly smooth function minimization [Monteiro and Svaiter, 2013, Gasnikov et al., 2018, Jiang et al., 2018, Bubeck et al., 2018].

We prove Theorem 7 by generalizing an acceleration framework due to [Monteiro and Svaiter, 2013]. This framework was recently used by several results to obtain near-optimal query complexities for minimizing highly smooth convex functions [Gasnikov et al., 2018, Jiang et al., 2018, Bubeck et al., 2018]. In Section B.1 we provide a variant of this general framework that is amenable to the noise induced by our oracles. In Section B.2 we show how to instantiate our framework using the oracles assuming a particular type of line search can be performed. In Section B.3 we then prove the Theorem 7. The algorithm for and analysis of line search is deferred to Appendix E.

## 3.2 Highly parallel optimization

With Theorem 7 in hand, to obtain our result we need to provide, for an appropriate function $\omega$, a highly parallel implementation of an approximate proximal step oracle and an approximate gradient oracle for a function that is an $O(\varepsilon)$ additive approximation $f$. As with previous work [Duchi et al., 2012, Scaman et al., 2018] we consider the convolution of $f$ with a Gaussian of covariance $r^2 \cdot I_d$ for $r > 0$ we will tune later. Formally, we define $g : \mathbb{R}^d \to \mathbb{R}$ for all $x \in \mathbb{R}^d$ as

$$g(x) \overset{\text{def}}{=} \int_{\mathbb{R}^d} \gamma_r(y) f(x - y) dy \quad \text{where} \quad \gamma_r(x) \overset{\text{def}}{=} \frac{1}{(\sqrt{2\pi}r)^d} \exp\left(-\frac{\|x\|^2}{2r^2}\right)$$

It is straightforward to prove (See Section D.1) the following standard facts regarding $g$.

**Lemma 8** *The function $g$ is convex, $L$-Lipschitz, and satisfies both $|g(y) - f(y)| \leq \sqrt{d} \cdot Lr$ and $\nabla^2 g(y) \preceq (L/r) \cdot I_d$ for all $y \in \mathbb{R}^d$.*

Consequently, to minimize $f$ up to $\varepsilon$ error, it suffices to minimize $g$ to $O(\varepsilon)$ error with $r = O(\frac{\varepsilon}{\sqrt{d}L})$. In the remainder of this section we simply show how to provide highly parallel implementations of the requisite oracles to achieve this by Theorem 7.

Now, as we have discussed, one way we could achieve this goal would be to use random sampling to approximate (in parallel) the $k$-th order Taylor approximation to $g$ and minimize a regularization of this function to implement the approximate proximal step oracle. While this procedure is depth-efficient, its work is quite large. Instead, we provide a more work efficient application of our acceleration framework. To implement a query to the oracles at some point $c \in \mathbb{R}^d$ we instead simply take multiple samples from $\gamma_r(x - c)$, i.e., the normal distribution with covariance $r^2 I_d$ and mean $c$, and use these samples to build an approximation to the gradient field of $g$. The algorithm for this procedure is given by Algorithm 1 and carefully combines the gradients of the sampled points to build a model with small bias and variance. By concentration bound and $\varepsilon$-net argument, we can show that Algorithm 1 outputs a vector field $v : \mathbb{R}^d \to \mathbb{R}^d$ that is an uniform approximation of $\nabla g$ within a small ball (See Section D.2 for the proof.)

---

**Algorithm 1:** Compute vector field approximating $\nabla g$

---

1 **Input**: Number of samples $N$, radius $r > 0$, error parameter $\eta \in (0, 1)$, center $c \in \mathbb{R}^d$.
2 Sample $x_1, x_2, \cdots, x_N$ independently from $\gamma_r(x - c)$.
3 **return** $v : \mathbb{R}^d \to \mathbb{R}^d$ defined for all $y \in \mathbb{R}^d$ by

$$v(y) = \frac{1}{N} \sum_{i=1}^{N} \frac{\gamma_r(y - x_i)}{\gamma_r(c - x_i)} \cdot \nabla f(x_i) \cdot \chi((x_i - c)^\top (y - c)) \cdot 1_{\|x_i - c\| \leq (\sqrt{d} + \frac{1}{\eta})r}$$

where $\chi(t) \overset{\text{def}}{=} 0$, if $|t| \geq r^2$, $\chi(t) \overset{\text{def}}{=} 1$ if $|t| \leq \frac{r^2}{2}$ and $\chi(t) \overset{\text{def}}{=} 2 - \frac{2|t|}{r^2}$ otherwise.

---

**Lemma 9 (Uniform Approximation)** *Algorithm 1 outputs vector field $v : \mathbb{R}^d \to \mathbb{R}$ such that for any $\delta \in (0, \frac{1}{2})$ with probability at least $1 - \delta$ the following holds*

$$\max_{y : \|y - c\| \leq \frac{\eta}{4}r} \|v(y) - \nabla g(y)\| \leq 5L \cdot \exp\left(-\frac{1}{2\eta^2}\right) + \frac{8L}{\sqrt{N}} \sqrt{\frac{d}{\eta^2} \log(9d) + \log \frac{1}{\delta}}.$$

*Consequently, for any $\varepsilon \in [0, 1]$, $N = \mathcal{O}([d \log d \log(\frac{1}{\varepsilon}) + \log(\frac{1}{\delta})]\varepsilon^{-2})$, and $\eta = \frac{1}{2\sqrt{\log\left(\frac{10}{\varepsilon}\right)}}$ this yields that $\max_{y : \|y - c\| \leq \widetilde{r}} \|v(y) - \nabla g(y)\| \leq L \cdot \varepsilon$ where $\widetilde{r} = \frac{r}{8\sqrt{\log\left(\frac{10}{\varepsilon}\right)}}$.*

This lemma immediately yields that we can use Algorithm 1 to implement a highly-parallel approximate gradient oracle for $g$. Interestingly, it can also be leveraged to implement a highly-parallel approximate proximal step oracle. Formally, we show how to use it to find $y$ such that

$$\nabla g(y) + \omega(\|y - x\|) \cdot (y - x) \approx 0 \quad \text{where} \quad \omega(s) \overset{\text{def}}{=} \frac{4Ls^p}{\widetilde{r}^{p+1}} \tag{5}$$

for some $p$ to be determined later. Ignoring logarithmic factors and supposing for simplicity that $L, R \leq 1$, Theorem 7 shows that by invoking this procedure $\widetilde{O}(k) \approx d^{\frac{p+1}{3p+4}} \varepsilon^{\frac{-2p-2}{3p+4}}$ times we could achieve function error on the order

$$\omega(1/k^{3/2})/k^2 \approx \widetilde{r}^{-(p+1)} k^{-\frac{3p+4}{2}} \approx d^{\frac{p+1}{2}} \varepsilon^{-(p+1)} k^{-\frac{3p+4}{2}} \approx \varepsilon$$

and therefore achieve the desired result by setting $p$ to be polylogarithmic in the problem parameters.

Consequently, we simply need to find $y$ satisfying (5). The algorithm that achieves this is Algorithm 2 which essentially performs gradient descent on

$$g(y) + \Phi(\|y - c\|) \quad \text{where} \quad \Phi(s) = \int_0^s \omega(t) \cdot t \, dt . \tag{6}$$

The performance of this algorithm is given by Theorem 24 in Section D.3. Combined with all of the above it proves Theorem 4, see Section D.4 for the details.

---

**Algorithm 2:** Approximate minimization of $g(y) + \Phi(\|y - c\|)$

---

1 **Input**: center $c \in \mathbb{R}^d$, accuracy $\varepsilon$, inner radius $\widetilde{r} = \frac{r}{8\sqrt{\log\left(\frac{60}{\varepsilon}\right)}}$, and step size $h = \frac{\widetilde{r}}{48p\sqrt{d}L}$.

2 Use Algorithm 1 to find a vector field $v$ such that $\max_{y:\|y-c\| \leq \widetilde{r}} \|v(y) - \nabla g(y)\| \leq L \cdot \frac{\varepsilon}{6}$.

3 $y \leftarrow c$.

4 **for** $i = 1, 2, \cdots \infty$ **do**

5 $\quad \delta_y = v(y) + \omega(\|y - c\|) \cdot (y - c)$ where $\omega$ is defined by (5) with $p \geq 1$.

6 $\quad$ **if** $\|\delta_y\| \leq L \cdot \frac{5\varepsilon}{6}$ **then return** $y$ **else** $y = y - h \cdot \delta_y$;

7 **end**

---

### Acknowledgments

The authors thank the anonymous reviewers for their helpful feedback in preparing this final version. Further, the authors are grateful for multiple funding sources which supported this work in part, including NSF Awards CCF-1740551, CCF-1749609, and DMS-1839116 and NSF CAREER Award CCF-1844855.

## Footnotes

[1]Throughout we assume that $f$ is differentiable, though our results carry over to the case where $f$ is non-differentiable and given by a sub-gradient oracle. This generalization is immediate as our analysis and algorithms are stable under finite-precision arithmetic and convex functions are almost everywhere differentiable.

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
