[Supplementary Material]

# A Further details on the lower bound

Our formal lower bound reads as follows:

**Theorem 9** *Let $\rho \in (0,1)$, and let $C = 12 + 4\log_d(Q/\rho)$. Assume that $\log(N)N\sqrt{\frac{C\log(d)}{d}} \leq \frac{1}{4}$ (i.e., $N \lesssim \sqrt{d/\log^3(d)}$). Fix a randomized algorithm that queries at most $Q$ points per iteration (both function value and gradient), and that runs for at most $N$ iterations. Then, with probability at least $1 - \rho$, when run on the shielded Nemirovski function $f$ one has for any queried point:*

$$f(x) - f^* \geq \frac{1}{4\sqrt{N}}.$$

To prove Theorem 9, we consider the following game between the algorithm (player A) issuing the queries, and the adversary (player B) building the hard *shielded Nemirovski* function $f$ (as defined in Section 2.3 and Section 2.4), i.e., player B chooses the orthonormal vectors in the definition of $f$. To make explicit the dependency of the shielded Nemirovski function on the choice of the orthonormal vectors $v_1, \cdots, v_N$, we denote it by $f^{v_1, \cdots, v_N}$ (with similar notation for the Nemirovski function $\mathcal{N}$ and the wall function $\mathcal{W}$). We restrict our attention to a deterministic player A and randomized player B, which is without loss of generality thanks to the minimax theorem. The game has $N$ iterations, and at each iteration $t$, players A and B maintain a common set of orthonormal vectors $\mathcal{V}_t = \{v_1, v_2, \cdots, v_t\}$, and common sets of vectors $\mathcal{Q}_1, \mathcal{Q}_2, \cdots, \mathcal{Q}_t$ where initially $\mathcal{Q}_0 = \emptyset$. At each iteration,

1. Simultaneously:

    (a) Player A queries a set of $Q$ points $\mathcal{Q}_t = \{z_t^{(1)}, \cdots, z_t^{(Q)}\}$ inside the unit ball.

    (b) Player B randomly sample $N - t + 1$ orthonormal vectors $v_t^{(t)}, v_t^{(t+1)}, \cdots, v_t^{(N)}$ from $\text{span}(\mathcal{V}_{t-1})^\perp$.

2. Player B returns $f^{v_1, v_2, \cdots, v_t, v_t^{(t+1)}, \cdots, v_t^{(N)}}(x)$ and $\nabla f^{v_1, v_2, \cdots, v_t, v_t^{(t+1)}, \cdots, v_t^{(N)}}(x)$ to player A for every $x \in \mathcal{Q}_t$ where $v_t := v_t^{(t)}$.

Note that at each iteration Player B answers the query with a different function, however we will show that in fact with high probability all the given answers are consistent with the final function. More precisely let us introduce the high probability event under which we will carry the proof. We say that Player B wins the game if the following holds:

$$\forall t \in [N], z \in \mathcal{Q}_t, s_1, s_2 \geq t, \left|\langle x, v_{s_1}^{(s_2)}\rangle\right| < \sqrt{\frac{C\log d}{d}} \cdot \left\|P_{V_{t-1}^\perp} x\right\|.$$

**Lemma 10** *Let $\rho \in (0,1)$. Assume $N \leq \frac{d}{2}$ and let $C = 12 + 4\log_d(Q/\rho)$. Then player B wins with probability at least $1 - \rho$.*

**Proof** For any $s_1$ and $s_2$, we note that $v_{s_1}^{(s_2)}$ follows the uniform distribution on the unit sphere restricted on the subspace $V_{s_1-1}^\perp$. For any $x \in \mathcal{Q}_t$, we have that

$$\langle x, v_{s_1}^{(s_2)}\rangle = \langle P_{V_{s_1-1}^\perp} x, P_{V_{s_1-1}^\perp} v_{s_1}^{(s_2)}\rangle.$$

By [Ball, 1997, Lemma 2.2], we have that

$$\mathbb{P}_{v_{s_1}^{(s_2)}}\left(\left|\langle P_{V_{s_1-1}^\perp} x, P_{V_{s_1-1}^\perp} v_{s_1}^{(s_2)}\rangle\right| \geq t \cdot \|P_{V_{s_1-1}^\perp} x\|_2\right) \leq 2\exp\left(-\dim V_{s_1-1}^\perp \cdot \frac{t^2}{2}\right).$$

Since $\dim V_{s_1-1}^\perp = d - s_1 + 1 \geq d - N \geq \frac{d}{2}$ and $\|P_{V_{s_1-1}^\perp} x\|_2 \leq \|P_{V_{t-1}^\perp} x\|_2$ (using $t \leq s_1$), we have that

$$\mathbb{P}\left(\left|\langle x, v_{s_1}^{(s_2)}\rangle\right| > \sqrt{\frac{C\log d}{d}} \cdot \left\|P_{V_{t-1}^\perp} x\right\|_2\right) \leq 2\exp\left(-\frac{d}{2} \cdot \frac{1}{2} \cdot \frac{C\log d}{d}\right) = 2d^{-\frac{C}{4}}.$$

Taking union bound over at most $N^2$ pairs of $v_i^{(j)}$ and $NQ$ many $x$, we have that player B wins with probability at least $1 - Q \cdot d^{3 - \frac{C}{4}}$, which concludes the proof. ∎

Next we show that if Player B wins the game, then indeed all answers are consistent with the final function.

**Lemma 11** *Assume player B wins the game and that $\gamma = 2\delta\sqrt{\frac{C\log(d)}{d}}$. Then, for all $t \in [N]$ and all $x \in \mathcal{Q}_t$, we have that*

$$f^{v_1, v_2, \cdots, v_t, v_t^{(t+1)}, \cdots, v_t^{(N)}}(x) = f^{v_1, v_2, \cdots, v_t, v_{t+1}, \cdots, v_N}(x) \tag{7}$$

*and that*

$$\nabla f^{v_1, v_2, \cdots, v_t, v_t^{(t+1)}, \cdots, v_t^{(N)}}(x) = \nabla f^{v_1, v_2, \cdots, v_t, v_{t+1}, \cdots, v_N}(x) \tag{8}$$

**Proof** Fix any $t \in [N]$ and any $x \in \mathcal{Q}_t$. Write $x = w + z$ with $w \in V_{t-1}$ and $z \in V_{t-1}^\perp$. Since player B wins, we have that

$$\left| \langle z, v_t^{(s)} \rangle \right| \le \sqrt{\frac{C\log d}{d}} \cdot \|z\| \,,$$

for all $s \ge t$. Lemma 2 shows that

$$\mathcal{W}^{v_1, v_2, \cdots, v_t, v_t^{(t+1)}, \cdots, v_t^{(N)}}(x) = \mathcal{W}^{v_1, v_2, \cdots, v_t, v_{t+1}, \cdots, v_N}(x) \,. \tag{9}$$

Moreover the equations following Lemma 2 show that (9) also holds for the function $f$ itself provided that $\|z\| \ge \delta$ (indeed, as discussed there if the argmax index in the definition of the Nemirovski function is attained at an index $\ge t$ then in fact $f(x) = \mathcal{W}(x)$, and otherwise the Nemirovski function value itself does not depend on $v_t^{(t+1)}, \cdots, v_t^{(N)}$).

Thus we only need to consider the case where $\|z\| \le \delta$. In this case we prove that (9) also holds for the Nemirovski function (and thus it also holds for $f$). Indeed for any $s > t$

$$\langle v_s, x \rangle - \gamma \cdot s = \langle v_s, z \rangle - \langle v_t, z \rangle + \langle v_t, x \rangle - \gamma \cdot s$$

$$\le 2\sqrt{\frac{C\log d}{d}} \cdot \delta + \langle v_t, x \rangle - \gamma \cdot s$$

$$\le \langle v_t, x \rangle - \gamma \cdot t \,,$$

where the last inequality uses that $\sqrt{\frac{C\log d}{d}} \le \frac{\gamma}{2\delta}$. This concludes the proof of (7). For (8) we simply note that (7) remains true for infinitesimal perturbations of $x$. ∎

Finally we show that no queried point could have a suboptimal gap smaller than $o(1/\sqrt{N})$.

**Lemma 12** *Assume player B wins and that $\log(N)N\sqrt{\frac{C\log(d)}{d}} \le \frac{1}{4}$. Then, for all $t \in [N]$ and all $x \in \mathcal{Q}_t$, we have that*

$$f(x) - f^* \ge \frac{1}{4\sqrt{N}} \,.$$

**Proof** First we claim that

$$f(x) - f^* \ge \frac{1}{\sqrt{N}} - \sqrt{\frac{C\log(d)}{d}} - \gamma N \,.$$

This follows from (1), Lemma 1, and the fact that:

$$f(x) \ge \mathcal{N}(x) \ge \langle v_N, x \rangle - \gamma N \,.$$

Next recall from Lemma 11 that we take $\gamma = 2\delta\sqrt{\frac{C\log(d)}{d}}$, and from Lemma 1 that $\frac{\delta}{\log_2(1/\delta)} = 4\sqrt{\frac{CN\log(d)}{d}} + \frac{1}{\sqrt{N}} \le \frac{2}{\sqrt{N}}$ where the inequality follows from the assumption on $N$. In particular we

have $\delta \le \frac{\log(N/2)}{\sqrt{N}}$. Thus:

$$f(x) - f^* \ge \frac{1}{\sqrt{N}} - \sqrt{\frac{C\log(d)}{d}}\left(1 + 2\log(N/2)\sqrt{N}\right) \ge \frac{1}{4\sqrt{N}}\,,$$

where the second inequality follows from the assumption on $N$. ∎

# B   Acceleration with Approximate Proximal Step Oracles

Here we provide the proofs associated with Section 3.1 and prove Theorem 6. Our proof is split into several parts. In Section B.1 we provide the acceleration framework we leverage, in Section B.2 we show how to instantiate the framework using our oracles, and in Section B.3 we then prove Theorem 6. This analysis relies on a line search result deferred to Appendix E.

## B.1   Framework

In this section we present the general acceleration framework based on Monteiro and Svaiter [2013] which we leverage to achieve our result. This acceleration framework is given by Algorithm 3 and is a noise-tolerant analog of the one present in Bubeck et al. [2018]. The framework maintains points $x_k$ and $y_k$ in each iteration $k$. To compute the next point, a careful convex combination of them is chosen, denote $\widetilde{x}_k$, and the next $y_{k+1}$ is chosen a point that has similar properties to the result of an approximate proximal step oracle and the next $x_{k+1}$ is then the result of moving from $x_k$ in the direction of $\nabla g(y_{k+1})$. Here we provide general results regarding the iterates in the general setting of Algorithm 3. In the next section we show how to implement the framework and ultimately bound the error.

---

**Algorithm 3:** Acceleration Framework

1 **Input:** $x_0 = y_0 = 0_d$, $\sigma \in (0,1)$, $A_0 = 0$, $K > 0$
2 **for** $k = 0, \dots, K-1$ **do**
3   $\quad$ Compute $\lambda_{k+1} > 0$ and $y_{k+1} \in \mathbb{R}^d$ such that for

$$a_{k+1} \overset{\text{def}}{=} \frac{1}{2}\left[\lambda_{k+1} + \sqrt{\lambda_{k+1}^2 + 4\lambda_{k+1}A_k}\right]\,,\ A_{k+1} \overset{\text{def}}{=} A_k + a_{k+1}\,,\ \widetilde{x}_k \overset{\text{def}}{=} \frac{A_k}{A_{k+1}}y_k + \frac{a_{k+1}}{A_{k+1}}x_k,$$

$\quad$ the following condition holds

$$\|\lambda_{k+1}\nabla g(y_{k+1}) + y_{k+1} - \widetilde{x}_k\| \le \sigma\|y_{k+1} - \widetilde{x}_k\| + \lambda_{k+1}\delta\,. \tag{10}$$

4   $\quad$ Compute $x_{k+1}$ such that the following holds

$$\|x_{k+1} - (x_k - a_{k+1}\nabla g(y_{k+1}))\| \le a_{k+1}\delta \tag{11}$$

5 **end**
6 **return** $y_K$

---

**Remark 13** *The definition of $a_{k+1}$ was chosen such that $\lambda_{k+1}A_{k+1} = a_{k+1}^2$. To see this, note that $a_{k+1}$ is a solution to $a_{k+1}^2 - \lambda_{k+1}a_{k+1} - \lambda_{k+1}A_k = 0$, which is equivalent as $A_{k+1} = A_k + a_k$.*

In the following theorem we give a general bound for the quality of the iterates in Algorithm 3.

**Theorem 14 (Framework Convergence)** *Algorithm 3 above gives for all $k \ge 1$ that*

$$A_k\left[g(y_k) - g^*\right] + \frac{1}{2}\|x_k - x^*\|^2 + \sum_{i \in [k]}\frac{(1-\sigma)A_i}{2\lambda_i}\|y_i - \widetilde{x}_{i-1}\|^2 \le \frac{1}{2}\|x^*\|^2 + \delta_k$$

*where*

$$\delta_k = \delta\sum_{i \in [k]} a_i\|x_i - x^*\| + \frac{\delta^2}{2(1-\sigma)}\sum_{i \in [k]} a_i^2\,.$$

**Proof** Let $\Delta_{k+1} \overset{\text{def}}{=} x_{k+1} - (x_k - a_{k+1}\nabla g(y_{k+1}))$, $r_k \overset{\text{def}}{=} \frac{1}{2}\|x_k - x^*\|^2$, and $\varepsilon_k \overset{\text{def}}{=} g(y_k) - g^*$ so

$$\frac{1}{2}\|x_{k+1} - x^* - \Delta_{k+1}\|^2 = r_k + a_{k+1}\nabla g(y_{k+1})^\top (x^* - x_k) + \frac{a_{k+1}^2}{2}\|\nabla g(y_{k+1})\|^2 .$$

Now, since

$$x_k = y_k + \frac{A_{k+1}}{a_{k+1}}(\widetilde{x}_k - y_k) = y_{k+1} + \frac{A_{k+1}}{a_{k+1}}(\widetilde{x}_k - y_{k+1}) + \frac{A_k}{a_{k+1}}(y_{k+1} - y_k)$$

and by convexity $g(z) \geq g(y_{k+1}) + \nabla g(y_{k+1})^\top (z - y_{k+1})$ for all $z$ we have

$$a_{k+1}\nabla g(y_{k+1})^\top (x^* - x_k) \leq A_{k+1}\nabla g(y_{k+1})^\top (y_{k+1} - \widetilde{x}_k) + A_k\varepsilon_k - A_{k+1}\varepsilon_{k+1} .$$

Combining these inequalities and applying Cauchy Schwarz yields

$$r_{k+1} = \frac{1}{2}\|x_{k+1} - x^* - \Delta_{k+1}\|^2 + \Delta_{k+1}^\top (x_{k+1} - x^* - \Delta_{k+1}) + \frac{1}{2}\|\Delta_{k+1}\|^2$$

$$\leq r_k + A_{k+1}\nabla g(y_{k+1})^\top (y_{k+1} - \widetilde{x}_k) + A_k\varepsilon_k - A_{k+1}\varepsilon_{k+1} + \frac{a_{k+1}^2}{2}\|\nabla g(y_{k+1})\|^2$$

$$+ \|\Delta_{k+1}\|\|x_{k+1} - x^*\|$$

Now rearranging (10) and applying $(a + b)^2 \leq (1 + t)a^2 + (1 + t^{-1})b^2$ for $t = \frac{1-\sigma}{\sigma}$ yields

$$2\lambda_{k+1}\nabla g(y_{k+1})^\top (y_{k+1} - \widetilde{x}_k) + \lambda_{k+1}^2\|\nabla g(y_{k+1})\|^2 \leq -(1-\sigma)\|y_{k+1} - \widetilde{x}_k\|^2 + (1-\sigma)^{-1}\lambda_{k+1}^2\delta^2$$

Combining with the facts that $\lambda_k A_k = a_k^2$ and $\|\Delta_{k+1}\| \leq a_{k+1}\delta$ yields

$$r_{k+1} + A_{k+1}\varepsilon_{k+1} + \frac{(1-\sigma)A_{k+1}}{2\lambda_{k+1}}\|y_{k+1} - \widetilde{x}_k\|^2 \leq r_k + A_k\varepsilon_k + a_{k+1}\delta\|x_{k+1} - x^*\| + \frac{\delta^2}{2(1-\sigma)}a_{k+1}^2$$

Summing over $k$ and using that $A_0 = 0$ and $x_0 = 0$ yields the result. ∎

Next we show that for sufficiently small $\delta$, the error in Theorem 14 is increased by only a constant factor. This will allow us to apply Theorem 14 when $\delta \neq 0$.

**Lemma 15 (Error Tolerance)** *Algorithm 3 with $\delta \leq c\sqrt{1-\sigma}\|x^*\|/A_K$ for some $c, K \geq 0$ gives that $\delta_k \leq c(1 + 3c)\|x^*\|^2$. Consequently, if $c \leq \frac{1}{4}$ then for all $k \in [K]$*

$$A_k\left[g(y_k) - g^*\right] + \frac{1}{2}\|x_k - x^*\|^2 + \sum_{i\in[k]} \frac{(1-\sigma)A_i}{2\lambda_i}\|y_i - \widetilde{x}_{i-1}\|^2 \leq \|x^*\|^2 \tag{12}$$

*In particular, this implies that taking $\delta \leq \frac{\|x^*\|}{\mu \cdot A_K}$ for $\mu \overset{\text{def}}{=} \frac{4\sqrt{2}}{\sqrt{1-\sigma}}$ then $\|x_k - x^*\| \leq 2\|x^*\|$. Furthermore, we have that either $g(y_k) \leq g^* + \varepsilon$ or $A_k \leq \frac{\|x^*\|^2}{\varepsilon}$.*

**Proof** Theorem 14, the assumption on $\delta$, $\sigma \in [0, 1)$ and $A_K = \sum_{i\in[K]} a_i$ yield that for all $k \in [K]$

$$\frac{1}{2}\|x_k - x^*\|^2 \leq \frac{1}{2}\|x^*\|^2 + c\|x^*\| \max_{i\in[K]} \|x_i - x^*\| + \frac{c^2}{2}\|x^*\|^2$$

Since this holds for all $k \in [K]$ it clearly holds for $k \in \arg\max_{i\in[K]} \|x_i - x^*\|$ and therefore

$$\max_{i\in[K]} \|x_i - x^*\|^2 - 2c\|x^*\| \max_{i\in[K]} \|x_i - x^*\| - (1 + c^2)\|x^*\|^2 \leq 0$$

Solving the quadratic and using $\sqrt{a + b} \leq \sqrt{a} + \sqrt{b}$ implies that

$$\max_{i\in[K]} \|x_i - x^*\| \leq \frac{1}{2}\left[2c\|x^*\| + \sqrt{4c^2\|x^*\|^2 + 4(1 + c^2)\|x^*\|^2}\right] \leq (c + (1 + c\sqrt{2}))\|x^*\| .$$

Therefore by the definition of $\delta_k$, we have

$$\delta_k = c[c + (1 + \sqrt{2}c)]\|x^*\|^2 + \frac{c^2}{2}\|x^*\|^2 \leq (3c^2 + c)\|x^*\|^2$$

for all $k \in [K]$ and (12) follows from Theorem 14 and that $c(1 + 3c) \leq \frac{1}{2}$ for $c \in [0, \frac{1}{4}]$. ∎

 ## B.2 Leveraging Approximate Proximal Step Oracle

Here we show how to implement and bound the convergence of Algorithm 3 given an approximate proximal step oracle. First, we show that given $\lambda_{k+1}\omega(\|y_{k+1} - \widetilde{x}_k\|)$ is sufficiently close to 1 then $y_{k+1}$ can be computed with an approximate proximal oracle. We show that such a $y_{k+1}$ can always be found (for suitable choice of $\sigma$) in Appendix E.

**Lemma 16 (Line Search Guarantee)** *If in each iteration $k$ of Algorithm 3 we choose $\lambda_{k+1}$ and $y_{k+1}$ such that for $d = \|y_{k+1} - \widetilde{x}_k\|$*

$$\|\nabla g(y_{k+1}) + \omega(d)(y_{k+1} - \widetilde{x}_k)\| \le \alpha \cdot \omega(d)d + \delta \quad and \quad \frac{1-\sigma}{1-\alpha} \le \lambda_{k+1}\omega(d) \le 1$$

*for $\alpha \in [0, 1)$ and $\omega : \mathbb{R}_+ \to \mathbb{R}_+$ then (10) is satisfied.*

**Proof** Leveraging that the assumptions imply $|\lambda_{k+1}\omega(d) - 1| = 1 - \lambda_{k+1}\omega(d)$ yields

$$\|\lambda_{k+1}\nabla g(y_{k+1}) + y_{k+1} - \widetilde{x}_k\| \le \lambda_{k+1}\|\nabla g(y_{k+1}) + \omega(d)(y_{k+1} - \widetilde{x}_k)\| + |\lambda_{k+1}\omega(d) - 1|\,\|y_{k+1} - \widetilde{x}_k\|$$
$$\le \lambda_{k+1}(\alpha \cdot \omega(d)d + \delta) + (1 - \lambda_{k+1}\omega(d))d$$
$$= [1 - (1 - \alpha)\lambda_{k+1}\omega(d)]d + \lambda_{k+1}\delta \, .$$

Since $(1 - \alpha)\lambda_{k+1}\omega(d) \ge 1 - \sigma$ by assumption the result follows. ∎

Note that the update $x_{k+1}$ can simply be read as $x_{k+1} = x_k - a_{k+1} \cdot v_{k+1}$ where $\|v_{k+1} - \nabla g(y_{k+1})\| \le \delta$. Consequently, $v_{k+1}$ can just be the result of a $\delta$-approximate gradient oracle (Definition 5). Consequently, this lemma shows that Algorithm 3 can be implemented with the oracles at our disposal, provided line search can be performed to achieve the guarantee of Lemma 16. We discuss this in the next section.

Next we bound the diameter of the iterates of the algorithm, i.e. how much the points vary.

**Lemma 17 (Diameter Bound)** *If in Algorithm 3 we have $\delta \le \frac{\|x^*\|}{\mu \cdot A_K}$ for $\mu \overset{\text{def}}{=} \frac{4\sqrt{2}}{\sqrt{1-\sigma}}$ and some $K > 0$. Then for all $k \in [K]$ and $\theta \in [0, 1$ we have $\|y_k - x^*\| \le \mu\|x^*\|$ and $\|\widetilde{x}_\theta - x^*\| \le \mu\|x^*\|$ for $\widetilde{x}_\theta = (1 - \theta)x_k + \theta y_k$.*

**Proof** Let $D_k = \|y_k - x^*\|$. Using $\widetilde{x}_k = \frac{A_k}{A_{k+1}}y_k + \frac{a_{k+1}}{A_{k+1}}x_k$, we have

$$\|\widetilde{x}_k - x^*\| \le \frac{A_k}{A_{k+1}}D_k + \frac{2a_{k+1}}{A_{k+1}}\|x^*\|.$$

Hence, $D_{k+1} \le \frac{A_k}{A_{k+1}}D_k + \frac{2a_{k+1}}{A_{k+1}}\|x^*\| + \|y_{k+1} - \widetilde{x}_k\|$. Rescaling and summing over $k$ yields

$$D_{k+1} \le 2\|x^*\| + \|y_{k+1} - \widetilde{x}_k\| + \frac{A_k}{A_{k+1}}\|y_k - \widetilde{x}_{k-1}\| + \frac{A_{k-1}}{A_{k+1}}\|y_{k-1} - \widetilde{x}_{k-2}\| + \cdots$$

$$\le 2\|x^*\| + \frac{1}{A_{k+1}}\sum_{j=1}^{k+1} A_j\|y_j - \widetilde{x}_{j-1}\|$$

$$\le 2\|x^*\| + \frac{\sqrt{\sum_{j=1}^{k+1} A_j\lambda_j}}{A_{k+1}}\sqrt{\sum_{j=1}^{k+1} \frac{A_j}{\lambda_j}\|y_j - \widetilde{x}_{j-1}\|^2}$$

$$\le 2\|x^*\| + \frac{\sqrt{\sum_{j=1}^{k+1} \lambda_j}}{\sqrt{A_{k+1}}}\sqrt{\frac{2\|x^*\|^2}{1-\sigma}}$$

$$\le 2\|x^*\| + \frac{2\sqrt{2}}{\sqrt{1-\sigma}}\|x^*\| \le \mu\|x^*\|$$

where we used $A_j$ is increasing and Lemma 15 in the third to last equation, and equation 14 for the second to last. The assumption on the relation between $\alpha$ and $\sigma$ implies $\sigma = \frac{1+\alpha}{2} = [\frac{1}{2}, 1)$ and the definition of $\mu$ gives the last inequality.

The second part of the claim follows by observing that $\widetilde{x}_\theta$ is a convex combination of $x_k$ and $y_k$, therefore

$$\|\widetilde{x}_\theta - x^*\| \le \max\{\|x_k - x^*\|, \|y_k - x^*\|\} \le \mu\|x^*\|.$$

∎

Finally, we bound the growth of $A_k$; this is crucial to derive the final convergence rate of the algorithm.

**Lemma 18 (Growth of $A_k$)** *Let* $\rho \stackrel{\text{def}}{=} \frac{1-\alpha}{1-\sigma} = 2$ *and* $\mu \stackrel{\text{def}}{=} \frac{4\sqrt{2}}{\sqrt{1-\sigma}} = \frac{8}{\sqrt{1-\alpha}}$. *If in Algorithm 3 for* $K \ge 0$ *we have* $\delta \le \frac{\|x^*\|}{\mu \cdot A_K}$ *and* $\lambda_k \ge \frac{1}{\rho \cdot \omega(\|y_k - \widetilde{x}_{k-1}\|)}$ *for all* $k \in \{0, ..., K\}$ *then for all* $J \in (0, \frac{k}{2})$ *we have*

$$A_k \ge \min\left\{ \frac{4^J}{\rho \cdot \omega(\mu\|x^*\|/4)} , \frac{(k/J)^2}{16\rho \cdot \omega\left(\frac{4\mu\|x^*\|}{(k/J)^{3/2}}\right)} \right\}.$$

*Further, if* $\|x^*\| \le R$, *for all* $k \in [K]$ *then* $A_k \ge \frac{1}{2\omega(2\mu R)}$.

**Proof** Let $d_k \stackrel{\text{def}}{=} \|y_k - \widetilde{x}_{k-1}\|$. By (12) of Lemma 15 we obtain for all $k \in [K]$

$$\sum_{i \in [k]} \frac{A_i}{\lambda_i} d_i^2 \le \frac{2\|x^*\|^2}{1-\sigma}. \tag{13}$$

Since $A_0 = 0$ we have $A_1 = a_1 = \lambda_1$ and consequently, (13) yields $d_1^2 \le \frac{2\|x^*\|^2}{1-\sigma}$ and therefore $d_1 \le \frac{\mu}{4}\|x^*\|$. Since $\omega$ is monotonic the assumptions imply

$$A_1 = \lambda_1 \ge \frac{1}{c \cdot \omega(d_1)} \ge \frac{1}{\rho \cdot \omega(\mu\|x^*\|/4)}.$$

Since the $A_k$ increase monotonically this immediately implies $A_k \ge A_1 \ge 1/[\rho\omega(\mu\|x^*\|/4)]$ as desired. Further, this implies that if $A_k \ge 4^J A_1$ then the result holds.

On the other hand, suppose $A_k < 4^J A_1$. Then, for some $1 \le i \le j \le k$ we have $A_j < 4A_i$ and $|j - i| \ge k/J$. The construction of $A_k$ then implies

$$\sqrt{A_j} > \sqrt{A_j} - \sqrt{A_i} = \sum_{t=i}^{j-1} \left[\sqrt{A_{t+1}} - \sqrt{A_t}\right] = \sum_{t=i}^{j-1} \frac{a_{t+1}}{\sqrt{A_{t+1}} + \sqrt{A_t}} \ge \frac{1}{2}\sum_{t=i}^{j-1} \sqrt{\lambda_{t+1}} \tag{14}$$

Hence, at least $\left\lceil \frac{j-i}{2} \right\rceil$ many $\lambda$'s have value less than $\frac{16A_j}{(j-i)^2}$. Letting $S$ denote the indices of these $\lambda$ we have by (13) that

$$\frac{2\|x^*\|^2}{1-\sigma} \ge \sum_{t \in S} \frac{A_t}{\lambda_t} d_t^2 \ge \left\lceil \frac{j-i}{2} \right\rceil \frac{A_i}{\left(\frac{16A_j}{(j-i)^2}\right)} \cdot \frac{1}{|S|}\sum_{t \in S} d_t^2 \ge \frac{(k/J)^3}{32 \cdot 4} \cdot \frac{1}{|S|}\sum_{t \in S} d_t^2$$

Consequently, $d_t \le \frac{16}{\sqrt{1-\sigma}} \frac{\|x^*\|}{(k/J)^{3/2}} \le \frac{4\mu\|x^*\|}{(k/J)^{3/2}}$ and $\lambda_t < \frac{16A_j}{(j-i)^2} \le \frac{16A_j}{(k/J)^2}$ for some $t \in [k]$. However, the monotonicity of $\omega$ and the assumptions on $\lambda$ also imply

$$\lambda_t \ge \frac{1}{\rho \cdot \omega(d_t)} \ge \frac{1}{\rho \cdot \omega\left(\frac{4\mu\|x^*\|}{(k/J)^{3/2}}\right)}$$

and the result now follows by observing that

$$A_k \ge A_t \ge \lambda_t \frac{(k/J)^2}{16}$$

giving the second term in the result. ∎

 **B.3 Putting It All Together**

Here we put together the analysis from the preceding sections and prove Theorem 6. Our proof relies on the following theorem giving our main guarantee regarding such a line search algorithm (See Section E for the proof.)

**Theorem 19 (Line Search Algorithm)** *Let $g : \mathbb{R}^d \to \mathbb{R}$ be a twice differentiable function that is minimized at a point $x^* \in \mathbb{R}^d$ with $\|x^*\| \le R$. Further, let $\omega : \mathbb{R}_+ \to \mathbb{R}_+$ be a continuously differentiable function where $0 < \omega'(s) \le \gamma \frac{\omega(s)}{s}$ for some fixed $\gamma \ge 1$ and all $s > 0$. Further, let $\mu \overset{\text{def}}{=} \frac{8}{\sqrt{1-\alpha}}$ and suppose*

$$\delta \le \min\left\{ \frac{\varepsilon}{\mu \cdot R \cdot 9c[(1+\alpha)c+1]} \;,\; 8\mu R \cdot \omega(8\mu R) \right\} \text{ and } 64\left(\alpha + \frac{1}{c}\right)\gamma^2 \le 1 \text{ for some } c \ge 1 \,.$$

*Then for any inputs $x^{(1)}, x^{(2)}$ with $\|x^{(1)}\|, \|x^{(2)}\| \le 2\mu R$, $\frac{1}{2\omega(2\mu R)} \le A \le \frac{R^2}{\varepsilon}$ there is an algorithm that returns $y$ and $\lambda$ such that $\widetilde{x} = \frac{a}{A+a}x^{(1)} + \frac{A}{A+a}x^{(2)}$ for $a = \frac{\lambda + \sqrt{\lambda^2 + 4\lambda A}}{2}$ that either satisfies*

$$g(y) \le g^* + \varepsilon \quad \text{and} \quad \omega(\|y - \widetilde{x}\|_2) \cdot \|y - \widetilde{x}\|_2 \le c \cdot \delta$$

*or, satisfies*

$$\frac{1}{2} \le \lambda \cdot \omega(\|y - \widetilde{x}\|_2) \le 1 \quad , \quad \omega(\|y - \widetilde{x}\|_2) \cdot \|y - \widetilde{x}\|_2 > c \cdot \delta,$$

*and*

$$\|\nabla g(y) + \omega(\|y - \widetilde{x}\|_2) \cdot (y - \widetilde{x})\| \le \alpha \cdot \omega(\|y - \widetilde{x}\|_2) \cdot \|y - \widetilde{x}\|_2 + \delta$$

*after*

$$6 + \log_2\left[ \left( \frac{160\mu R c}{\delta} + \frac{9R^2}{\varepsilon} \right) \cdot \omega(8c\mu R) \right]$$

*calls to the $(\alpha, \delta)$-approximate $\omega$-proximal step oracle $\mathcal{T}_{\text{prox}}$ for $g$.*

Leveraging this we can prove our main theorem regarding our acceleration framework. We first give this result below as a slightly more general result and then use it to immediately improve the theorem.

**Theorem 20 (General Tunable Acceleration Framework)** *Let $g : \mathbb{R}^d \to \mathbb{R}$ be a convex twice-differentiable function minimized at $x^*$ with $\|x^*\| \le R$, $\varepsilon > 0$, $\alpha \in [0, 1)$, and $c \ge 150, \gamma \ge 1$ such that $64(\alpha + c^{-1})\gamma^2 \le 1$. Further, let $\omega : \mathbb{R}_+ \to \mathbb{R}_+$ be a monotonically increasing continuously differentiable function with $0 < \omega'(s) \le \gamma \cdot \omega(s)/s$ for all $s > 0$. There is an algorithm which for all $k$ computes a point $y_k$ with*

$$g(y_k) - g^* \le \max\left\{ \varepsilon \;,\; \frac{32 \cdot \omega\left( \frac{4\mu\|x^*\|}{k^{3/2}} \right) \|x^*\|^2}{k^2} \right\} \quad \text{where} \quad \mu \overset{\text{def}}{=} \frac{8}{\sqrt{1-\alpha}}$$

*using $k(6 + \log_2[(1500\mu^3 R^3 c^2[(1+\alpha)c+1]) \cdot \omega(8c\mu R) \cdot \varepsilon^{-1}])^2$ queries to a $(\alpha, \delta)$-approximate $\omega$-proximal step oracle for $g$ and a $\delta$-approximate gradient oracle for $g$ provided that it holds that $\delta \le \varepsilon/[20\mu^3 R[(1+\alpha)c+1]]$ and $\varepsilon \le 72c[(1+\alpha)c+1](\mu R)^3 \cdot \omega(8\mu R)$.*

**Proof** Consider an application of Algorithm 3 where in each iteration $k$ we invoke Theorem 19 with $x^{(1)} = y_k$, $x^{(2)} = x_k$, and $A = A_k$ to compute $y_{k+1} = y$ and $\lambda_k = \lambda$. Now supposing that $A_k \le R^2/\varepsilon$ and that in this invocation we choose the $\delta$ of Theorem 19 to be $\delta' \overset{\text{def}}{=} \min\{\varepsilon'/(\mu R), 8\mu R \cdot \omega(8\mu R)\} = \varepsilon'/(\mu R)$ for $\varepsilon' \overset{\text{def}}{=} \varepsilon/[\mu \cdot R \cdot 9c[(1+\alpha)c+1]]$ , we have that the conditions of Lemma 15 and Theorem 19 are met as $\varepsilon' \le \varepsilon$. Further, if $\omega(\|y - \widetilde{x}\|_2) \cdot \|y - \widetilde{x}\|_2 \le c \cdot \delta'$ then we output $y_{k+1}$ and are guaranteed that $g(y_{k+1}) \le g^* + \varepsilon$ by Theorem 19 and the choice of parameters.

Otherwise, $\omega(\|y - \widetilde{x}\|_2) \cdot \|y - \widetilde{x}\|_2 > c \cdot \delta'$ and the necessary conditions are met for Algorithm 3 to proceed by Lemma 16. Further, in this case, we have that

$$\lambda_{k+1} \le \frac{1}{\omega(\|y_{k+1} - \widetilde{x}_k\|)} \le \frac{\|y_{k+1} - \widetilde{x}_k\|}{c \cdot \delta'} \le \frac{2\mu\|x^*\|}{c \cdot \delta'}.$$

Furthermore, the assumption that $A_k \leq \frac{\|x^*\|^2}{\varepsilon}$, Remark 13, and the assumption on $\delta$ yield

$$A_{k+1} = A_k + a_{k+1} \leq A_k + \sqrt{A_{k+1}} \cdot \sqrt{\frac{2\mu\|x^*\|}{c \cdot \delta'}} \leq \frac{\|x^*\|^2}{\varepsilon} + \frac{1}{2}A_{k+1} + \frac{\mu\|x^*\|}{c\delta'}$$

which implies that

$$A_{k+1} \leq \frac{2\|x^*\|^2}{\varepsilon} + \frac{2\mu\|x^*\|}{c \cdot \delta'} \leq \frac{2R^2}{\varepsilon} + \frac{19\mu^2 R^2[(1+\alpha)c+1]}{\varepsilon} \leq \frac{20\mu^2 R^2[(1+\alpha)c+1]}{\varepsilon}$$

Since, $\|x^*\|/(\mu A_{k+1}) \geq \varepsilon/[20\mu^3 R[(1+\alpha)c+1]] \geq \delta$ by the assumption $c \geq 150$, we have that Lemma 15 still holds and therefore either $A_{k+1} \leq \|x^*\|^2/\varepsilon$ or $g(y_{k+1}) - g^* \leq \varepsilon$ and we can repeat the inductive argument.

Consequently, if after $k$ steps we have not already returned an $\varepsilon$-approximate point then we have from Lemma 15 and Lemma 18 the convergence rate to an $\varepsilon$-optimal point of the general framework as

$$g(y_k) - g^* \leq \frac{\|x^*\|^2}{A_k} \leq \min_{J \in [\frac{k}{2}]} \max \left\{ \frac{2 \cdot \omega(\mu\|x^*\|/4)}{4^J}, \frac{32 \cdot \omega\left(\frac{4\mu\|x^*\|}{(k/J)^{3/2}}\right)}{(k/J)^2} \right\} \|x^*\|^2$$

and the convergence rate follows by considering $J = \lceil 1 + \log_4(2\|x^*\|^2\omega(\mu\|x^*\|/4)/\varepsilon)\rceil$ and the monotonicity of $\omega$. Putting together with Theorem 19, we have that for

$$\mathcal{K} \stackrel{\text{def}}{=} \left\lceil 1 + \log_4\left(\frac{2\|x^*\|^2\omega(\mu\|x^*\|/4)}{\varepsilon}\right)\right\rceil \cdot \left(6 + \log_2\left[\left(\frac{160\mu\|x^*\|c}{\delta'} + \frac{9\|x^*\|^2}{\varepsilon}\right) \cdot \omega(8c\mu\|x^*\|)\right]\right)$$

$$\leq \left(6 + \log_2\left[\frac{170\mu^2 R^2 c}{\varepsilon'} \cdot \omega(8c\mu R)\right]\right) \cdot \left\lceil 1 + \frac{1}{2}\log_2\left(2R^2\frac{\omega(\frac{\mu R}{4})}{\varepsilon}\right)\right\rceil$$

$$\leq \left(6 + \log_2\left[\frac{170\mu^2 R^2 c}{\varepsilon'} \cdot \omega(8c\mu R)\right]\right)^2 \leq \left(6 + \log_2\left[\frac{1500\mu^3 R^3 c^2[(1+\alpha)c+1]}{\varepsilon} \cdot \omega(8c\mu R)\right]\right)^2$$

$\mathcal{K}$ queries to a $(\alpha, \delta')$-approximate $\omega$-proximal step oracle is needed at each iteration.

∎

Leveraging this, we prove Theorem 6.

**Proof** [Proof of Theorem 6] Consider invoke Theorem 20 with $c = 150\gamma^2$. Since $\gamma \geq 1$ we have $c \geq 150$. Further, since $\alpha \leq 1/(128\gamma^2)$ and $c^{-1} \leq 1/(128\gamma^2)$ we have $64(\alpha + c^{-1})\gamma^2 \leq 1$. Further, under these assumptions we have $\mu \stackrel{\text{def}}{=} 8/(\sqrt{1-\alpha}) \leq 10$ and $[(1+\alpha)c+1] \leq 200\gamma^2$. Consequently, $\delta$ and $\varepsilon$ are constrained sufficiently to invoke Theorem 20 and the result follows. ∎

# C  Applications

Here we briefly sketch several applications of the acceleration framework described in Section 3.1. First we show how minimizing the regularized $p$-th order Taylor approximation to $g$ is yields an approximate $\omega$-proximal step oracle.

**Lemma 21 (Accelerated Taylor Descent)** *Suppose that $\nabla^p g$ is $L_p$-Lipschitz and that $\mathcal{T}(x) \stackrel{\text{def}}{=} \arg\min_y g_p(y; x) + \frac{L_p+L}{p!}\|y-x\|^{p+1}$ where $g_p(y; x)$ is the value of the $p$'th order Taylor approximation of $g$ about $x$ evaluated at $y$ and $L \geq 0$. Then, $\mathcal{T}_{\text{prox}}$ is a $((1+p)^{-1}(1+L/L_p)^{-1}, 0)$-approximate $\omega$-proximal step oracle (Definition 4) for $\omega(d) \stackrel{\text{def}}{=} \frac{(L_p+L) \cdot (p+1)}{p!}d^{p-1}$.*

**Proof** Let $y = \mathcal{T}_{\text{prox}}(x)$ for arbitrary $x$. The optimality conditions of $y$ yield that

$$\nabla_y g_p(y; x) = \frac{(p+1)(L_p + L)}{p!}\|y - x\|^{p-1}(x - y) = \omega(\|y - x\|)(x - y).$$

Further, since Taylor expansion of $\nabla g(y)$ yields

$$\|\nabla g(y) + \omega(\|y - x\|)(y - x)\| = \|\nabla g(y) - \nabla_y g_p(y; x)\| \leq \frac{L_p}{p!}\|y - x\|^p$$

$$= \frac{L_p}{(1 + p)(L_p + L)}\omega(\|y - x\|)\|y - x\|$$

the result follows by observing that $\alpha = (1 + p)^{-1}(1 + L/L_p)^{-1}$ and $\delta = 0$, as claimed. ∎

Now, note that for $\omega(d)$ defined in this lemma we have that $\omega'(d) = (p - 2)\omega(s)/s$. Consequently, with respect to Theorem 6 we have that $\gamma = p - 2$ and $\alpha = (1 + p)^{-1}(1 + L/L_p)^{-1}$ for the oracle defined in this lemma. Consequently, by picking $L = O(L_p\text{poly}(p))$ this oracle satisfies the necessary conditions of the theorems and therefore (up to logarithmic factors) with $k$ queries to the oracle and a gradient oracle invoking Theorem 6 yields that one can compute a point $y_k$ with

$$g(y_k) - g^* \lesssim \frac{\omega(\frac{\|x^*\|}{k^{3/2}})\|x^*\|^2}{k^2} \lesssim \frac{(L_p + L) \cdot (p + 1) \cdot \|x^*\|^{p+1}}{p! \cdot k^{\frac{3p+1}{2}}} \, .$$

This matches the rate of [Gasnikov et al., 2018, Jiang et al., 2018, Bubeck et al., 2018] up to polylogarithmic factors.

Next we show how approximately minimizing a regularization of $g$ yields an approximate $\omega$-proximal step oracle.

**Lemma 22 (Approximate Proximal Point)** *Suppose that $g$ is $L$-smooth and convex and that $\mathcal{T}(x)$ is a point $y_x$ where for $G_x(y) \overset{\text{def}}{=} g(y) + \frac{\kappa}{2}\|y - x\|^2$ we have $G_x(y_x) - G_x^* \leq \rho$ where $G_x^*$ is the minimum value of $G_x$. Then, $\mathcal{T}_{\text{prox}}$ is a $(0, \rho(L + \kappa))$-approximate $\omega$-proximal step oracle (Definition 4) for $\omega(d) \overset{\text{def}}{=} \kappa$.*

**Proof** Since $G$ is $L + \kappa$-smooth we have that

$$\rho \geq \frac{1}{L + \kappa}\|\nabla G_x(y_x)\| = \frac{1}{L + \kappa}\|\nabla g(y_x) + \kappa(y_x - x)\| \, .$$

The result follows by observing that $\alpha = 0$ and $\delta = \rho(L + \kappa)$, as claimed. ∎

Now, note that for $\omega(d)$ defined in this lemma we have that $\omega'(d) = 0$. Consequently, with respect to Theorem 6 we have that $\gamma = 0$ and $\alpha = 0$ for the oracle defined in this lemma. Consequently, this oracle satisfies the necessary conditions of the theorems for some $\varepsilon$ so long as $\rho = O(\varepsilon/[\|x^*\|(L+\kappa)]$ and therefore (up to logarithmic factors) with $k$ queries to the oracle and a gradient oracle invoking Theorem 6 yields that one can compute a point $y_k$ with

$$g(y_k) - g^* \lesssim \frac{\omega(\frac{\|x^*\|}{k^{3/2}})\|x^*\|^2}{k^2} \lesssim \frac{\kappa \cdot \|x^*\|^2}{k^2} \, .$$

This matches the rate of [Frostig et al., Lin et al., 2015] up to polylogarithmic factors with slightly stronger assumptions. We leave it to future work to use this framework to fully generalize this result and develop further applications.

# D Upper Bound

Here we provide the proofs associated with Section 3.2 and prove Theorem 3. Our proof is split into several parts. In Section D.1 we provide basic facts about the convolved function we optimize, in Section D.2 we analyze our algorithm for approximating the gradient, in Section D.3 we then analyze our algorithm for computing an approximate proximal step, and in Section D.4 we then put everything together to prove Theorem 3.

Throughout this section we use $\|\cdot\|_{op}$ to denote the operator norm of a matrix and $D$ as the differential operator.

## D.1 Gaussian Convolution for Approximation

Here we prove Lemma 7 which provides basic facts about $g$, e.g. convexity and continuity, that we use throughout our analysis.

**Proof** [Proof of Lemma 7] Since $g$ is a weighted linear combination of shifted $f$, i.e.

$$g(y) = \int_{\mathbb{R}^d} \gamma_r(x) f(y - x) dx$$

and as $f$ is convex, so is $g$. Similarly, we have $g$ is $L$-Lipschitz. Finally, we note that

$$|g(y) - f(y)| \leq \int_{\mathbb{R}^d} \gamma_r(y-x)|f(x) - f(y)|dx \leq L \int_{\mathbb{R}^d} \gamma_r(y-x)\|x-y\|_2 dx = L \cdot \mathbb{E}_{x\sim\gamma_r}\|x\|_2 \leq L\sqrt{d} \cdot r$$

where we used $\mathbb{E}_{x\sim\gamma_r}\|x\|_2 \leq \sqrt{\mathbb{E}_{x\sim\gamma_r}\|x\|_2^2} \leq \sqrt{d} \cdot r$.

Next, we note that $\nabla g = \gamma_r * \nabla f$ and hence $\nabla^2 g = \nabla \gamma_r * \nabla f$

$$v^\top \nabla^2 g(y) v = \int_{\mathbb{R}^d} \gamma_r(y-x) \cdot \left\langle -\frac{y-x}{r^2}, v \right\rangle \cdot \langle \nabla f(x), v \rangle \, dy.$$

So we have for any $\|v\|_2 = 1$, by the fact that $f$ is $L$-Lipschitz that

$$\left| v^\top \nabla^2 g(y) v \right| \leq \frac{L}{r} \cdot \int_{\mathbb{R}^d} \gamma_r(y-x) \left| \left\langle \frac{y-x}{r}, v \right\rangle \right| dy = \frac{L}{r} \cdot \mathbb{E}_{\zeta\sim\mathcal{N}(0,1)}|\zeta| = \frac{L}{r} \cdot \sqrt{\frac{2}{\pi}} \leq \frac{L}{r}.$$

and therefore $\|\nabla^2 g(y)\|_{\mathrm{op}} \leq \frac{L}{r}$. ∎

## D.2 Noisy Gradient Oracle: Sampling

In this section we prove Lemma 8 bounding the performance of Algorithm 1 for approximating the gradient of $g$. We begin by studying each sampled vector in Algorithm 1.

**Lemma 23 (Statistics of one sample)** *Given a $L$-Lipschitz function $f$ on $\mathbb{R}^d$, a vector $c$, radius $r > 0$, and error parameter $1 > \eta > 0$. Sample $x$ according to $\gamma_r(x - c)$. Define the vector field*

$$\ell(y) \stackrel{\mathrm{def}}{=} \frac{\gamma_r(y-x)}{\gamma_r(c-x)} \cdot \nabla f(x) \cdot \chi((x-c)^\top(y-c)) \cdot 1_{\|x-c\|\leq(\sqrt{d}+\frac{1}{\eta})r}.$$

*For any $y$ such that $\|y - c\| \leq \frac{\eta}{4}r$, we have that*

$$\|\mathbb{E}\ell(y) - \nabla(\gamma_r * f)(y)\|_2 \leq 2L \cdot \exp(-\frac{1}{2\eta^2}),$$

$$\|\ell(y)\|_2 \leq 3L,$$

$$\|D\ell(y)\|_{\mathrm{op}} \leq \frac{20L\sqrt{d}}{r\eta}.$$

**Proof** For the bias, we note that

$$\mathbb{E}\ell(y) = \int_{\mathbb{R}^d} \frac{\gamma_r(y-x)}{\gamma_r(x-c)} \cdot \nabla f(x) \cdot \chi((x-c)^\top(y-c)) \cdot \gamma_r(x-c) \cdot 1_{\|x-c\|\leq(\sqrt{d}+\frac{1}{\eta})r} \, dx$$

$$= \int_{\mathbb{R}^d} \gamma_r(y-x) \cdot \nabla f(x) \cdot \chi((x-c)^\top(y-c)) \cdot 1_{\|x-c\|\leq(\sqrt{d}+\frac{1}{\eta})r} \, dx$$

$$= \nabla(\gamma_r * f)(y) - \int_{\mathbb{R}^d} \gamma_r(y-x) \cdot \nabla f(x) \cdot \beta(y, x) \, dx$$

where

$$\beta(y, x) = 1 - \chi((x-c)^\top(y-c)) \cdot 1_{\|x-c\|\leq(\sqrt{d}+\frac{1}{\eta})r}.$$

Since $1 \geq \beta(y, x) \geq 0$ for all $x, y$, we have

$$\|\mathbb{E}\ell(y) - \nabla(\gamma_r * f)(y)\|_2 \leq \int_{\mathbb{R}^d} \gamma_r(y - x) \cdot \|\nabla f(x)\|_2 \cdot \beta(y, x) \, dx$$

$$\leq L \cdot \int_{\mathbb{R}^d} \gamma_r(y - x) \cdot \beta(y, x) \, dx$$

$$\leq L \cdot \mathbb{P}_x \left[ \beta(y, x) > 0 \right].$$

Now, we note that $\beta(y, x) > 0$ implies either $\|x - c\| > (\sqrt{d} + \frac{1}{\eta})r$ or $|(x - c)^\top(y - c)| > \frac{r^2}{2}$.

By a tail bound of Chi-square distribution [Laurent and Massart, 2000], we have

$$\mathbb{P}_x \left( \|x - c\| > \left( \sqrt{d} + \frac{1}{\eta} \right) r \right) \leq \exp \left( -\frac{1}{2\eta^2} \right). \tag{15}$$

Next, we note that for any fixed $c$ and $y$, $(x - c)^\top(y - c)$ follows the normal distribution $\mathcal{N}(\|y - c\|^2, \|y - c\|^2 r^2)$ when $x$ is sampled from $\gamma_r(y - x)$. By the assumption that $\|y - c\| \leq \frac{\eta}{4}r \leq \frac{r}{4}$, we have that

$$\mathbb{P}_x \left[ |(x - c)^\top(y - c)| > \frac{r^2}{2} \right] \leq \mathbb{P}_{\zeta \sim \mathcal{N}(0,1)} \left( |\zeta| \geq \frac{r^2}{4\|y - c\|r} \right)$$

$$\leq \exp \left( -\frac{r^2}{32\|y - c\|^2} \right)$$

$$\leq \exp(-\frac{1}{2\eta^2}). \tag{16}$$

Union bound over case (15) and case (16) gives that $\mathbb{P}_x \left[ \beta(y, x) > 0 \right] \leq 2\exp(-\frac{1}{2\eta^2})$. This gives the bound on $\mathbb{E}\ell(y)$.

For the bound on $\|\ell\|$, we note from the Lipschitz assumption of $f$ that

$$\|\ell(y)\|_2 \leq L \cdot \frac{\gamma_r(y - x)}{\gamma_r(x - c)} \cdot \chi((x - c)^\top(y - c)).$$

$$\leq L \cdot \frac{\gamma_r(y - x)}{\gamma_r(x - c)} \cdot 1_{|(x - c)^\top(y - c)| \leq r^2}$$

For any $x$ with $|(x - c)^\top(y - c)| \leq r^2$, we have that

$$\log \frac{\gamma_r(y - x)}{\gamma_r(x - c)} = -\frac{1}{2r^2}\|y - x\|_2^2 + \frac{1}{2r^2}\|c - x\|_2^2$$

$$= \frac{1}{2r^2} \left( -2(c - x)^\top(y - c) - \|y - c\|_2^2 \right)$$

$$\leq \frac{|(x - c)^\top(y - c)|}{r^2} < 1. \tag{17}$$

Hence, we have $\|\ell(y)\|_2 \leq 3L$.

For the bound of the Jacobian of $\ell$, we note that

$$D\ell(y) = \frac{\gamma_r(y - x)}{\gamma_r(x - c)} \cdot \nabla f(x) \cdot \left( -\frac{y - x}{r^2} \right)^\top \cdot \chi((x - c)^\top(y - c)) \cdot 1_{\|x - c\| \leq (\sqrt{d} + \frac{1}{\eta})r}$$

$$+ \frac{\gamma_r(y - x)}{\gamma_r(c - x)} \cdot \nabla f(x) \cdot (x - c)^\top \chi'((x - c)^\top(y - c)) \cdot 1_{\|x - c\| \leq (\sqrt{d} + \frac{1}{\eta})r}.$$

Since the Lipschitz constant of $\chi$ is bounded by $\frac{2}{r^2}$ and the Lipschitz assumption of $f$ is bounded by $L$, (17) and the above equation shows that

$$\|D\ell(y)\|_{\mathrm{op}} \leq e \cdot L \cdot \frac{\|y - x\|_2}{r^2} \cdot 1_{\|x - c\| \leq (\sqrt{d} + \frac{1}{\eta})r} + e \cdot L \cdot \|x - c\|_2 \cdot \frac{2}{r^2} \cdot 1_{\|x - c\| \leq (\sqrt{d} + \frac{1}{\eta})r}$$

$$\leq e \cdot L \cdot \frac{(\sqrt{d} + \frac{1}{\eta} + \frac{\eta}{4})r}{r^2} + 2e \cdot L \frac{(\sqrt{d} + \frac{1}{\eta})r}{r^2}$$

$$\leq \frac{Lr}{r^2} + \frac{9L}{r} \cdot (\sqrt{d} + \frac{1}{\eta}) \leq \frac{20L\sqrt{d}}{r\eta}.$$

By a concentration and $\varepsilon$-net argument we use Lemma 23 to prove Lemma 8.

**Proof** [Proof of Lemma 8] Fix $y$ such that $\|y - c\|_2 \leq \frac{\eta}{4}r$, we let $v(y) - \nabla g(y) = \frac{1}{N} \sum_{i=1}^{N} \varepsilon^{(i)}$ be the sum of $N$ independent vectors $\varepsilon^{(i)}$. Lemma 23 shows that

$$\|\mathbb{E}\varepsilon^{(i)}\|_2 \leq 2L \cdot \exp\left(-\frac{1}{2\eta^2}\right) \text{ and } \|\varepsilon^{(i)}\|_2 \leq 3L + L = 4L.$$

Pinelis's inequality [Pinelis, 1994] shows that

$$\mathbb{P}\left(\left\|\frac{1}{N}\sum_{i=1}^{N}\varepsilon^{(i)}\right\|_2 \geq 2L \cdot \exp(-\frac{1}{2\eta^2}) + 4L \cdot t\right) \leq 2\exp\left(-\frac{Nt^2}{2}\right).$$

To make this holds for all $y$ with $\|y - c\|_2 \leq \frac{\eta}{4}r$, we pick an $\varepsilon$-net $\mathcal{N}$ on $\{y : \|y - c\|_2 \leq \frac{\eta}{4}r\}$ with $\varepsilon = \frac{\eta}{4}r \cdot \frac{\exp(-\frac{1}{2\eta^2})}{3\sqrt{d}}$. It is known that $|\mathcal{N}_\varepsilon(B_d(0, r))| \leq (\frac{3r}{\varepsilon})^d$, therefore, using $0 < \eta \leq 1$

$$|\mathcal{N}| \leq \left(\frac{3\frac{\eta}{4}r}{\frac{\eta}{4}r \cdot \frac{\exp(-\frac{1}{2\eta^2})}{3\sqrt{d}}}\right)^d = \left(9\sqrt{d}\right)^d \exp\left(\frac{d}{2\eta^2}\right) \leq \exp\left(\frac{d\log(81d)}{\eta^2}\right).$$

For any $y$ with $\|y - c\|_2 \leq \frac{\eta}{4}r$, there is $y' \in \mathcal{N}$ with $\|y' - y\|_2 \leq \varepsilon$, therefore by Lemma 7 we have

$$\|\nabla g(y') - \nabla g(y)\| \leq \frac{L\varepsilon}{r} \leq L \cdot \exp(-\frac{1}{2\eta^2}).$$

Lemma 23 shows that $\|Dv(y)\|_{\text{op}} \leq \frac{20L\sqrt{d}}{r\eta}$. Hence, we have

$$\|v(y') - v(y)\|_2 \leq \varepsilon \cdot \frac{20L\sqrt{d}}{r\eta} \leq 2L\exp(-\frac{1}{2\eta^2}).$$

Taking the union bound on $\mathcal{N}$, we have that

$$\mathbb{P}\left(\max_{y:\|y-c\|_2 \leq \frac{\eta}{4}r} \|v(y) - \nabla g(y)\|_2 \geq 5L \cdot \exp(-\frac{1}{2\eta^2}) + 4L \cdot t\right) \leq 2\exp\left(\frac{d\log(81d)}{\eta^2} - \frac{Nt^2}{2}\right).$$

Setting $2\exp\left(\frac{d\log(81d)}{\eta^2} - \frac{Nt^2}{2}\right) = \delta$, we get

$$4L \cdot t \leq \frac{4L}{\sqrt{N}}\sqrt{\frac{2d\log(81d)}{\eta^2} + 2\log\frac{2}{\delta}} \leq \frac{8L}{\sqrt{N}}\sqrt{\frac{d\log(9d)}{\eta^2} + \log\frac{1}{\delta}}$$

on the LHS.

### D.3 Approximate Proximal Step Oracle Implementation

Here we prove the following theorem which bounds the performance of Algorithm 2.

**Theorem 24** *Algorithm 2 outputs $y$ such that $\|\nabla g(y) + \omega(\|y - c\|) \cdot (y - c)\| \leq L \cdot \varepsilon$ in $\mathcal{O}(\frac{p\sqrt{d}}{\varepsilon^2})$ iterations with $N = \mathcal{O}([d\log d \log(\frac{1}{\varepsilon}) + \log(\frac{1}{\delta})]\varepsilon^{-2})$ oracle calls to $f$ in parallel with probability at least $1 - \delta$ where $\omega$ is defined by (5) with $\widetilde{r} = \frac{r}{8\sqrt{\log(\frac{60}{\varepsilon})}}$.*

**Proof** [Proof of Theorem 24] First, we need to prove that $y$ stays inside $\|y - c\|_2 \leq \widetilde{r}$. Given this, the correctness of the output follows from the error bound on $v$ and the stopping condition.

We prove $\|y - c\|_2 \leq \widetilde{r}$ by induction. Let $y'$ be the one step from $y$, namely $y' = y - h \cdot \delta_y$. Then, we have

$$\|y' - c\|_2 \leq \|y - h \cdot \omega(\|y - c\|_2) \cdot (y - c) - c\|_2 + h\|v(y)\|_2$$

$$= |1 - h \cdot \omega(\|y - c\|_2)| \, \|y - c\|_2 + \frac{4}{3}Lh$$

where we used the induction hypothesis $\|y - c\|_2 \leq \widetilde{r}$ and the approximation guarantee to show that $\|v(y)\|_2 \leq \|v(y) - \nabla g(y)\|_2 + \|\nabla g(y)\|_2 \leq \frac{L\varepsilon}{6} + L \leq \frac{4}{3}L$. Next, we note from the assumption on step size that

$$h \cdot \omega(\|y - c\|_2) \leq h \cdot \frac{4L\widetilde{r}^p}{\widetilde{r}^{p+1}} \leq 1.$$

Hence, we have

$$\|y' - c\|_2 \leq (1 - h \cdot \omega(\|y - c\|_2)) \, \|y - c\|_2 + \frac{4}{3}Lh$$

$$= \|y - c\|_2 - h\omega(\|y - c\|_2)\|y - c\|_2 + \frac{4}{3}Lh.$$

Note that $\frac{4}{3}Lh \leq \frac{\widetilde{r}}{3p}$. Hence, if $\|y - c\|_2 \leq \left(1 - \frac{1}{3p}\right)\widetilde{r}$, we know that $\|y' - c\|_2 \leq \widetilde{r}$. Otherwise if $\|y - c\|_2 \geq \left(1 - \frac{1}{3p}\right)\widetilde{r}$, we know that

$$\omega(\|y - c\|_2)\|y - c\|_2 \geq \frac{4L}{\widetilde{r}^{p+1}}\left(1 - \frac{1}{3p}\right)^p \widetilde{r}^{p+1} \geq \frac{4}{3}L$$

which implies $\|y' - c\|_2 \leq \|y - c\|_2$. Hence, in both cases, we have $\|y' - c\|_2 \leq \widetilde{r}$. This completes the induction.

Finally, we need to bound the number of iterations before the algorithm terminates. Let $\mathcal{L}(y) := g(y) + \Phi(\|y - c\|_2)$ where $\Phi$ is defined in (6). By Lemma 7, we have that

$$\nabla^2 \mathcal{L} \preceq \left(\frac{L}{r} + \frac{5L\sqrt{d}}{\widetilde{r}}\right) \cdot I_d \preceq \frac{6L\sqrt{d}}{\widetilde{r}} \cdot I_d$$

Hence, by smoothness we have

$$\mathcal{L}(y') \leq \mathcal{L}(y) - h\langle\nabla\mathcal{L}(y), \delta_y\rangle + 3\frac{L}{\widetilde{r}}\sqrt{d} \cdot h^2\|\delta_y\|^2.$$

Note that $\delta_y = \nabla\mathcal{L}(y) + \eta$ for some vector $\eta$ such that $\|\eta\|_2 \leq L \cdot \frac{\varepsilon}{6}$ by the approximation guarantee. Therefore

$$\mathcal{L}(y') \leq \mathcal{L}(y) - h\|\nabla\mathcal{L}(y)\|^2 + h\|\nabla\mathcal{L}(y)\|\|\eta\| + 3\frac{L}{\widetilde{r}}\sqrt{d}h^2(2\|\nabla\mathcal{L}(y)\|^2 + 2\|\eta\|^2)$$

$$\leq \mathcal{L}(y) - \frac{7h}{8}\|\nabla\mathcal{L}(y)\|^2 + h\|\nabla\mathcal{L}(y)\|\|\eta\| + \frac{h}{8}\|\eta\|^2$$

$$\leq \mathcal{L}(y) - \frac{7h}{8}\|\nabla\mathcal{L}(y)\|^2 + \frac{h}{2}\|\nabla\mathcal{L}(y)\|^2 + \frac{h}{2}\|\eta\|^2 + \frac{h}{8}\|\eta\|^2$$

$$\leq \mathcal{L}(y) - \frac{7h}{8}\left(L \cdot \frac{2\varepsilon}{3}\right)^2 + \frac{5h}{8}\left(L \cdot \frac{\varepsilon}{6}\right)^2$$

$$= \mathcal{L}(y) - \frac{h}{3}L^2\varepsilon^2 = \mathcal{L}(y) - \frac{\widetilde{r}L\varepsilon^2}{144p\sqrt{d}}$$

where we used that $\|\nabla\mathcal{L}(y)\| \geq \|\delta_y\| - \|\eta\| \geq \frac{2\varepsilon}{3}L$ from the stopping criteria. This shows that $\mathcal{L}$ decreased by $\frac{\widetilde{r}L\varepsilon^2}{144p\sqrt{d}}$ every iteration. Since $\mathcal{L}$ has Lipschitz constant $L + 4L = 5L$ on $\|y - c\| \leq \widetilde{r}$,

$$\max_{\|y-c\|\leq\widetilde{r}}\mathcal{L}(y) - \min_{\|y-c\|\leq\widetilde{r}}\mathcal{L}(y) \leq 10L\widetilde{r}.$$

Therefore the number of step is at most $\mathcal{O}(\frac{p\sqrt{d}}{\varepsilon^2})$ and we have

$$\|\nabla g(y) + \omega(\|y - c\|_2) \cdot (y - c)\|_2 \leq \frac{L\varepsilon}{6} + \frac{5\varepsilon}{6}L \leq L \cdot \varepsilon$$

678   as claimed.                                                                                     ∎

679

680   The above theorem shows that we can implement (1) a noisy gradient oracle with $\beta = \frac{L \cdot \varepsilon}{6}$; and (2)
681   an optimization oracle with $\alpha = 0$ and $\delta = L \cdot \varepsilon$. Since by Theorem 24 we have $\|y_{k+1} - \widetilde{x}_k\| \leq \widetilde{r}$,
682   i.e., the output of the optimization oracle is bounded in a ball of radius $\widetilde{r}$ from the center, therefore
683   $g_{y_{k+1}} := v(y_{k+1})$ as the vector field formed by sampling satisfies $\|g_{y_{k+1}} - \nabla g(y_{k+1})\| \leq \frac{\delta}{6}$,
684   justifying its validity as a noisy gradient oracle at $y_{k+1}$.

### D.4   Parallel Complexity

686   Here we show how to put everything together to prove Theorem 3, our main highly-parallel optimiza-
687   tion result.

688   **Proof** [Proof of Theorem 3] Invoking the result of Section B and following the discussion in
689   Section D.1, with $r = \frac{\varepsilon}{\sqrt{d}L}$, we have $\widetilde{r} = \frac{r}{\sqrt{\log(\frac{60}{\varepsilon'})}} = \frac{\varepsilon}{L\sqrt{d \log(\frac{60}{\varepsilon'})}}$ and since

$$\omega(x) = \frac{4Lx^p}{\widetilde{r}^{p+1}} = \frac{4L^{p+2}x^p[d\log(\frac{60}{\varepsilon'})]^{\frac{p+1}{2}}}{\varepsilon^{p+1}},$$

690   from Theorem 6 we have for $\frac{\gamma^2}{c} = \frac{p^2}{c} \leq \frac{1}{64}$, the convergence rate to an $\varepsilon$-optimal point as

$$f(y_k) - f^* = \mathcal{O}\Big(\frac{\omega(\frac{\|x^*\|}{k^{3/2}})}{k^2}\|x^*\|^2\Big) = \mathcal{O}\Big(\frac{L^{p+2}\|x^*\|^p[d\log(\frac{1}{\varepsilon'})]^{\frac{p+1}{2}}}{\varepsilon^{p+1} \cdot k^2 \cdot k^{\frac{3p}{2}}}\|x^*\|^2\Big)$$

691   with $\mathcal{O}\Big(\frac{d\log d\log(\frac{1}{\varepsilon'}) + \log(\frac{1}{\rho})}{\varepsilon'^2} \times \mathcal{K}\Big)$ (sub)gradient queries to $f$ in parallel in each round for $\varepsilon' = $
692   $\mathcal{O}(\frac{\varepsilon}{\|x^*\| \cdot L})$, as required by the accuracy for which the optimization oracle is implemented in Theo-
693   rem 6 and the number of proximal oracle calls the line search procedure needs where

$$\mathcal{K} \overset{\text{def}}{=} \Big(6 + \log_2\Big[\frac{1500\mu^3 R^3 c^2[(1+\alpha)c+1]}{\varepsilon}\omega(8c\mu R)\Big]\Big)^2 = \mathcal{O}\Big(\log^2\Big[\frac{L^{p+2}\|x^*\|^{p+3}[d\log(\frac{1}{\varepsilon'})]^{\frac{p+1}{2}}}{\varepsilon^{p+2}}\Big]\Big).$$

694   Setting the result to the desired accuracy $\varepsilon$, we have that it suffices to pick $k = K$ for

$$K = \mathcal{O}\Big(\Big[L^{p+2} \cdot \|x^*\|^{p+2}\Big]^{\frac{2}{3p+4}} \cdot \Big[\frac{[d\log(\frac{\|x^*\| \cdot L}{\varepsilon})]^{\frac{p+1}{2}}}{\varepsilon^{p+2}}\Big]^{\frac{2}{3p+4}}\Big)$$

$$= \mathcal{O}\Big(\Big[L^{p+2} \cdot \|x^*\|^{p+2}\Big]^{\frac{2}{3p+4}} \cdot \Big(\frac{d}{\varepsilon^2}\Big)^{\frac{p+1}{3p+4}}\Big(\frac{1}{\varepsilon}\Big)^{\frac{2}{3p+4}} \cdot \Big[\log\Big(\frac{\|x^*\| \cdot L}{\varepsilon}\Big)\Big]^{\frac{p+1}{3p+4}}\Big)$$

695   Picking $p$ such that $\log(\frac{d}{\varepsilon^2}) = 3(3p + 4)$, end up with

$$K = \mathcal{O}\Big(\Big(\frac{d}{\varepsilon^2}\Big)^{\frac{1}{3}} \cdot \Big(\frac{1}{\varepsilon}\Big)^{\frac{1}{\log(d/\varepsilon^2)}} \cdot \log^{\frac{1}{3}}\Big(\frac{1}{\varepsilon}\Big) \cdot \Big(\log\Big(\frac{1}{\varepsilon}\Big)\Big)^{\frac{1}{\log(d/\varepsilon^2)}}\Big)$$

696   which is $\widetilde{\mathcal{O}}(d^{1/3}\varepsilon^{-2/3})$, as claimed. Setting $\rho = \mathcal{O}(\frac{\nu}{K})$ for the algorithm to succeed with probability
697   at least $1 - \nu$, denote $\eta \overset{\text{def}}{=} \log(\frac{d}{\varepsilon^2})$ the number of parallel (sub)gradient queries is

$$\mathcal{O}\Big(\frac{d\log d\log(\frac{1}{\varepsilon}) + \log(d^{1/3}\varepsilon^{-2/3}/\nu)}{\varepsilon^2} \times \mathcal{K}\Big)$$

$$= \mathcal{O}\Big(\frac{d\log d\log(\frac{1}{\varepsilon}) + \log(d^{1/3}\varepsilon^{-2/3}/\nu)}{\varepsilon^2} \times \log^2\Big[\frac{[d\log(\frac{1}{\varepsilon})]^{\frac{p+1}{2}}}{\varepsilon^{p+2}}\Big]\Big)$$

$$= \mathcal{O}\Big(\frac{d\log d\log(\frac{1}{\varepsilon}) + \log(d^{1/3}\varepsilon^{-2/3}/\nu)}{\varepsilon^2} \times \log^2\Big[\frac{[d\log(\frac{1}{\varepsilon})]^{\frac{1}{18}\eta - \frac{1}{6}}}{\varepsilon^{\frac{1}{9}\eta + \frac{2}{3}}}\Big]\Big)$$

698   With the choice of $p$, it suffices to pick $c$ large enough such that $\frac{81c}{64} \geq (\log(\frac{d}{\varepsilon^2}) - 12)^2$ for the
699   assumption to hold.                                                                             ∎

700

# E   Line Search Implementation

In this section, we assume access to an $(\alpha, \delta)$-approximate $\omega$-proximal step oracle $\mathcal{T}_{\text{prox}}$ for a convex function $g$. The goal is to use $\mathcal{T}_{\text{prox}}$ to find a point $y$ that satisfies Lemma 16, as required by the algorithm framework at each iteration. In particular, below is the assumption we are making and the main theorem we are going to prove, which we recall from Appendix B.

**Theorem 19 (Line Search Algorithm)** *Let $g : \mathbb{R}^d \to \mathbb{R}$ be a twice differentiable function that is minimized at a point $x^* \in \mathbb{R}^d$ with $\|x^*\| \leq R$. Further, let $\omega : \mathbb{R}_+ \to \mathbb{R}_+$ be a continuously differentiable function where $0 < \omega'(s) \leq \gamma \frac{\omega(s)}{s}$ for some fixed $\gamma \geq 1$ and all $s > 0$. Further, let $\mu \overset{\text{def}}{=} \frac{8}{\sqrt{1-\alpha}}$ and suppose*

$$\delta \leq \min\left\{ \frac{\varepsilon}{\mu \cdot R \cdot 9c[(1+\alpha)c + 1]} , \, 8\mu R \cdot \omega(8\mu R) \right\} \text{ and } 64\left(\alpha + \frac{1}{c}\right)\gamma^2 \leq 1 \text{ for some } c \geq 1 .$$

*Then for any inputs $x^{(1)}, x^{(2)}$ with $\|x^{(1)}\|, \|x^{(2)}\| \leq 2\mu R$, $\frac{1}{2\omega(2\mu R)} \leq A \leq \frac{R^2}{\varepsilon}$ there is an algorithm that returns $y$ and $\lambda$ such that $\widetilde{x} = \frac{a}{A+a}x^{(1)} + \frac{A}{A+a}x^{(2)}$ for $a = \frac{\lambda + \sqrt{\lambda^2 + 4\lambda A}}{2}$ that either satisfies*

$$g(y) \leq g^* + \varepsilon \quad and \quad \omega(\|y - \widetilde{x}\|_2) \cdot \|y - \widetilde{x}\|_2 \leq c \cdot \delta$$

*or, satisfies*

$$\frac{1}{2} \leq \lambda \cdot \omega(\|y - \widetilde{x}\|_2) \leq 1 \quad , \quad \omega(\|y - \widetilde{x}\|_2) \cdot \|y - \widetilde{x}\|_2 > c \cdot \delta,$$

*and*

$$\|\nabla g(y) + \omega(\|y - \widetilde{x}\|_2) \cdot (y - \widetilde{x})\| \leq \alpha \cdot \omega(\|y - \widetilde{x}\|_2) \cdot \|y - \widetilde{x}\|_2 + \delta$$

*after*

$$6 + \log_2\left[ \left( \frac{160\mu Rc}{\delta} + \frac{9R^2}{\varepsilon} \right) \cdot \omega(8c\mu R) \right]$$

*calls to the $(\alpha, \delta)$-approximate $\omega$-proximal step oracle $\mathcal{T}_{\text{prox}}$ for $g$.*

We assume $\delta \leq \frac{\varepsilon'}{\mu \cdot R}$ to make sure the oracle gives out information for different $x$ (and therefore we can achieve sufficiently small error). Furthermore, assume $\delta \leq 8\mu R \cdot \omega(8\mu R)$. The reason is that if both $x$ and $y$ lie in a radius $\mu R$ ball, $\alpha \cdot \omega(\|y - x\|_2) \cdot \|y - x\|_2$ is bounded by $2\mu R \cdot \omega(2\mu R)$. So if $\delta$ is much larger than this, the oracle essentially can always output the same $y$ regardless of $x$.

## E.1   Line Search Algorithm

To simplify the notation, we define $\widetilde{x}_\theta \overset{\text{def}}{=} (1 - \theta)x^{(1)} + \theta x^{(2)}$. Now, our goal is to find $\theta$ such that

$$\frac{1}{2} \leq \zeta(\theta) \leq 1 \quad \text{where} \quad \zeta(\theta) \overset{\text{def}}{=} \lambda_\theta \cdot \omega(\|y_\theta - \widetilde{x}_\theta\|_2) \tag{18}$$

for $y_\theta = \mathcal{T}_{\text{prox}}(\widetilde{x}_\theta)$ and $\lambda_\theta = \frac{(1-\theta)^2 A}{\theta}$.

First, we note that $\zeta(0) = +\infty$ and $\zeta(1) = 0$ (or otherwise, we find an approximate minimizer).

**Lemma 25** *We have either $\zeta(0) = +\infty$ or $g(x^{(1)}) \leq g(x^*) + \varepsilon'$. Moreover, we have $\zeta(1) = 0$.*

**Proof** By the definition of the $(\alpha, \delta)$ proximal oracle, we have

$$\left\| \nabla g(\mathcal{T}_{\text{prox}}(x^{(1)})) + \omega(\|\mathcal{T}_{\text{prox}}(x^{(1)}) - x^{(1)}\|) \cdot (\mathcal{T}_{\text{prox}}(x^{(1)}) - x^{(1)}) \right\|$$

$$\leq \alpha \cdot \omega(\|\mathcal{T}_{\text{prox}}(x^{(1)}) - x^{(1)}\|) \cdot \|\mathcal{T}_{\text{prox}}(x^{(1)}) - x^{(1)}\| + \delta.$$

If $\|\mathcal{T}_{\text{prox}}(x^{(1)}) - x^{(1)}\| = 0$, we have $\mathcal{T}_{\text{prox}}(x^{(1)}) = x^{(1)}$ and hence $\|\nabla g(x^{(1)})\| \leq \delta$. By convexity of $g$, we have that from Lemma 17

$$g(x^{(1)}) \leq g(x^*) + \delta\|x^{(1)} - x^*\|_2 \leq g(x^*) + \mu\delta R \leq g(x^*) + \varepsilon'.$$

where we used $\delta \leq \frac{\varepsilon'}{\mu \cdot R}$ at the end. Otherwise, we have $\|\mathcal{T}_{\text{prox}}(x^{(1)}) - x^{(1)}\| > 0$ therefore $\zeta(0) = +\infty$ and $\zeta(1) = 0$ from the definition. ∎

Therefore, to find $\theta$ such that $\zeta(\theta) = \frac{3}{4}$, we can simply perform binary search. In particular, in $\log_2(\frac{1}{\tau})$ iterations, we can find $0 \leq \ell \leq u \leq 1$ with $|\ell - u| \leq \tau$ such that $\zeta(\ell) - \frac{3}{4}$ and $\zeta(u) - \frac{3}{4}$ have different signs. See Algorithm 4 for the algorithm details. The key question is how small $\tau$ we need to make sure $\frac{1}{2} \leq \zeta(\frac{\ell+u}{2}) \leq 1$.

The difficultly here is that $\zeta$ may not be continuous. Therefore, we cannot bound the Lipschitz constant of $\zeta$ directly. Different from previous papers [Bubeck et al., 2018], our proof does not depend on how we implement the proximal oracle $\mathcal{T}_{\text{prox}}$ and do not assume how $\mathcal{T}_{\text{prox}}(x)$ changes with respect to $x$. In fact, the oracle $\mathcal{T}_{\text{prox}}$ we constructed in Section D may not even give the same output for the same input. Therefore, it is difficult to bound how far $\mathcal{T}_{\text{prox}}(x)$ changes under the change of $\lambda$. To avoid this problem, we first relate the noisy oracle $\mathcal{T}_{\text{prox}}$ with the ideal oracle with $\alpha = \delta = 0$. We note that the ideal oracle is exactly performing a proximal step as follows:

**Lemma 26 (Exact Proximal Map)** *Given $x$, let $y^* := \mathcal{O}(x) := \arg\min_y G(y)$ where*

$$G(y) \stackrel{\text{def}}{=} g(y) + W(\|y - x\|_2) \quad \text{with} \quad W(s) \stackrel{\text{def}}{=} \int_0^s \omega(u) \cdot u \, du$$

*then $\mathcal{O}$ is a $(0,0)$ proximal oracle for $g$. Furthermore, $G$ is strictly convex with $\nabla^2 G(y) \succeq \omega(\|y - x\|_2) \cdot I$ for any $x$.*

**Proof** From the optimality condition we have for $y^* = \mathcal{O}(x)$

$$\nabla G(y^*) = \nabla g(y^*) + \omega(\|y^* - x\|_2) \cdot (y^* - x) = 0.$$

which means $\mathcal{O}$ is a $(0,0)$ proximal oracle according to the definition. Note that

$$\nabla^2 G(y) = \nabla^2 g(y) + \omega(\|y - x\|_2)I + \omega'(\|y - x\|_2) \cdot \frac{(y - x)(y - x)^\top}{\|y - x\|_2}$$
$$\succeq \omega(\|y - x\|_2)I$$

where we used $g$ is convex and $\omega$ is increasing. Since $G$ is strictly convex, this shows that $y^*$ is the unique minimizer of $G$. ∎

In Section E.2, we show that $\zeta$ is close to some continuous function $\zeta^*$ (except for some cases that we can handle separately).

## E.2  Line Search Regime: Relation between Exact and Inexact Proximal Map

The goal of this section is to relate

$$\zeta(\theta) \stackrel{\text{def}}{=} \frac{(1-\theta)^2 A}{\theta} \omega(\|y_\theta - \widetilde{x}_\theta\|_2)$$

for $y_\theta = \mathcal{T}_{\text{prox}}(\widetilde{x}_\theta)$ is output of an $(\alpha, \delta)$ proximal oracle to

$$\zeta^*(\theta) = \frac{(1-\theta)^2 A}{\theta} \omega(\|y_\theta^* - \widetilde{x}_\theta\|_2)$$

where $y_\theta^* = \arg\min_y G_\theta(y)$ with

$$G_\theta(y) = g(y) + W(\|y - \widetilde{x}_\theta\|_2), \tag{19}$$

the exact proximal map. In particular, we will show in Lemma 28 that $\zeta(\theta)$ is an constant approximation of $\zeta^*(\theta)$. Therefore, one can study the binary search of $\zeta$ via $\zeta^*$.

First we give a lemma that relates $\|y_\theta - \widetilde{x}_\theta\|_2$ and $\|y_\theta^* - \widetilde{x}_\theta\|_2$.

---

**Algorithm 4:** Line Search Algorithm

---

**1 Input**: $x^{(1)}, x^{(2)} \in \mathbb{R}^d$ and $\frac{1}{2\omega(2\mu R)} \leq A \leq \frac{R^2}{\varepsilon}$.

**2 Input:** $\varepsilon' = \frac{\varepsilon}{9c((1+\alpha)c+1)} \in (0,1]$.

**3 Input**: an $(\alpha, \delta)$ proximal oracle $\mathcal{T}_{\text{prox}}$ for a convex twice-differentiable function $g$.

**4 Assumption:** $\|x^{(1)}\|_2 \leq 2\mu R$, $\|x^{(2)}\|_2 \leq 2\mu R$, $\|x^*\|_2 \leq R$ for some minimizer $x^*$ of $g$.

**5 Assumption:** $\delta \leq \min\{\frac{\varepsilon'}{\mu \cdot R}, 8\mu R \cdot \omega(8\mu R)\}$. $0 < \omega'(s) \leq \gamma \frac{\omega(s)}{s}$ for all $s > 0$. $\frac{1-\sigma}{1-\alpha} = \frac{1}{2}$. $64(\alpha + \frac{1}{c})\gamma^2 \leq 1$ for some $c \geq 1$.

**6** Define $\widetilde{x}_\theta = (1-\theta)x^{(1)} + \theta x^{(2)}$, $y_\theta = \mathcal{T}_{\text{prox}}(\widetilde{x}_\theta)$ and $\zeta(\theta)$ according to (18).

**7** Let $\tau = \min\left\{\frac{1}{4}, \frac{1}{2}\sqrt{\frac{1}{4}\frac{1}{A \cdot \omega(8c\mu R)}}, \frac{A\delta}{64\mu R}, \frac{c\delta}{360\mu\gamma R \cdot \omega(8c\mu R)}, \frac{1}{200\left(1+A \cdot \omega(8c\mu R) + \frac{4\mu R}{A\delta} + \frac{\mu R}{\delta} \cdot \omega(8c\mu R)\right)}\right\}$.

**8** Set $\ell = 0$, $u = 1$.

**9 while** $u \geq \ell + \tau$ **do**

**10**     $m = \frac{\ell + u}{2}$.

**11**     **if** $\zeta(m) \geq \frac{3}{4}$ **then**

**12**        $\ell \leftarrow m$.

**13**     **else**

**14**        $u \leftarrow m$.

**15**     **end**

**16 end**

**17 if** $\omega(\|y_\ell - \widetilde{x}_\ell\|_2) \cdot \|y_\ell - \widetilde{x}_\ell\|_2 \leq c \cdot \delta$ **then**

**18**     **Return** $y_\ell$ as an approximate minimizer.

**19 else if** $\omega(\|y_u - \widetilde{x}_u\|_2) \cdot \|y_u - \widetilde{x}_u\|_2 \leq c \cdot \delta$ **then**

**20**     **Return** $y_u$ as an approximate minimizer.

**21 else**

**22**     **Return** $y_\ell$ as an approximate solution for the line search.

**23 end**

---

**Lemma 27** *Assume that $8(\alpha + \frac{1}{c})\gamma \leq 1$. If $\omega(\|y_\theta - \widetilde{x}_\theta\|_2) \cdot \|y_\theta - \widetilde{x}_\theta\|_2 \geq c \cdot \delta$, then*

$$\left(1 - 8\left(\alpha + \frac{1}{c}\right)\gamma\right)\|y_\theta - \widetilde{x}_\theta\|_2 \leq \|y_\theta^* - \widetilde{x}_\theta\|_2 \leq \left(1 + 8\left(\alpha + \frac{1}{c}\right)\gamma\right)\|y_\theta - \widetilde{x}_\theta\|_2.$$

**Proof** We define $y_\theta^{(t)} = (1-t)y_\theta + ty_\theta^*$. Then, we have that

$$\nabla G_\theta(y_\theta^*) - \nabla G_\theta(y_\theta) = \int_0^1 \nabla^2 G_\theta(y_\theta^{(t)}) \cdot (y_\theta^* - y_\theta)dt. \tag{20}$$

Lemma 26 shows that

$$\nabla^2 G_\theta(y_\theta^{(t)}) \succeq \omega(\|y_\theta^{(t)} - \widetilde{x}_\theta\|_2) \cdot I. \tag{21}$$

To lower bound $\|y_\theta^{(t)} - \widetilde{x}_\theta\|_2$, we split the proof into two cases:

Case 1: $\|y_\theta - y_\theta^*\|_2 \geq 4\|y_\theta - \widetilde{x}_\theta\|_2$. Since $y_\theta^{(t)} = (1-t)y_\theta + ty_\theta^*$, then for $t \geq \frac{1}{2}$,

$$\|y_\theta^{(t)} - \widetilde{x}_\theta\|_2 = \|y_\theta - \widetilde{x}_\theta + t(y_\theta^* - y_\theta)\|_2$$
$$\geq t\|y_\theta^* - y_\theta\|_2 - \|y_\theta - \widetilde{x}_\theta\|_2$$
$$\geq \|y_\theta - \widetilde{x}_\theta\|_2.$$

Since $\omega$ is increasing, we have $\omega(\|y_\theta^{(t)} - \widetilde{x}_\theta\|_2) \geq \omega(\|y_\theta - \widetilde{x}_\theta\|_2)$. Together with (20) and (21), we have that

$$\|\nabla G_\theta(y_\theta) - \nabla G_\theta(y_\theta^*)\|_2 \geq \int_{1/2}^1 \omega(\|y_\theta - \widetilde{x}_\theta\|_2)dt \cdot \|y_\theta - y_\theta^*\|_2$$

$$= \frac{1}{2}\omega(\|y_\theta - \widetilde{x}_\theta\|_2) \cdot \|y_\theta - y_\theta^*\|_2.$$

Case 2: $\|y_\theta - y_\theta^*\|_2 \le 4\|y_\theta - \widetilde{x}_\theta\|_2$. Since $y_\theta^{(t)} = (1-t)y_\theta + ty_\theta^*$, we have

$$\|y_\theta^{(t)} - \widetilde{x}_\theta\|_2 \ge \|y_\theta - \widetilde{x}_\theta\|_2 - t\|y_\theta^* - y_\theta\|_2 \ge (1-4t)\|y_\theta - \widetilde{x}_\theta\|_2.$$

Using this and $\omega(\eta \cdot \beta) \le \eta^\gamma \omega(\beta)$ (which is implied by $\omega'(s) \le \gamma \frac{\omega(s)}{s}$ from Grönwall's inequality), for $0 \le t \le \frac{1}{4}$, we have that

$$\omega(\|y_\theta^{(t)} - \widetilde{x}_\theta\|_2) \ge (1-4t)^\gamma \omega(\|y_\theta - \widetilde{x}_\theta\|_2).$$

Together with (20) and (21), we have that

$$\|\nabla G_\theta(y_\theta) - \nabla G_\theta(y_\theta^*)\|_2 \ge \int_0^{1/4} (1-4t)^\gamma dt \cdot \omega(\|y_\theta - \widetilde{x}_\theta\|_2) \cdot \|y_\theta - y_\theta^*\|_2$$

$$= \frac{1}{4(\gamma+1)} \cdot \omega(\|y_\theta - \widetilde{x}_\theta\|_2) \cdot \|y_\theta - y_\theta^*\|_2. \quad (22)$$

In both cases, we have (22) as $\gamma \ge 1$.

On the other hand, the assumption on $y_\theta$ shows that

$$\|\nabla G_\theta(y_\theta) - \nabla G_\theta(y_\theta^*)\|_2 = \|\nabla G_\theta(y_\theta)\|_2 \le \alpha \cdot \omega(\|y_\theta - \widetilde{x}_\theta\|_2) \cdot \|y_\theta - \widetilde{x}_\theta\|_2 + \delta$$

$$\le (\alpha + \frac{1}{c}) \cdot \omega(\|y_\theta - \widetilde{x}_\theta\|_2) \cdot \|y_\theta - \widetilde{x}_\theta\|_2 \quad (23)$$

where we used the assumption on $\omega(\|y_\theta - \widetilde{x}_\theta\|_2) \cdot \|y_\theta - \widetilde{x}_\theta\|_2$.

Combining (22) and (23), we have that

$$\|y_\theta - y_\theta^*\|_2 \le 4(\alpha + \frac{1}{c})(\gamma+1) \cdot \|y_\theta - \widetilde{x}_\theta\|_2 \le 8(\alpha + \frac{1}{c})\gamma \cdot \|y_\theta - \widetilde{x}_\theta\|_2$$

where we used that $\gamma \ge 1$. The claim now follows from triangle inequality. ∎

Since $\zeta$ is only a function of $\|y_\theta - \widetilde{x}_\theta\|_2$, we have the following main result of this section:

**Lemma 28** *If* $64(\alpha + \frac{1}{c})\gamma^2 \le 1$ *and* $\omega(\|y_\theta - \widetilde{x}_\theta\|_2) \cdot \|y_\theta - \widetilde{x}_\theta\|_2 \ge c \cdot \delta$, *then* $\frac{7}{8}\zeta(\theta) \le \zeta^*(\theta) \le \frac{5}{4}\zeta(\theta)$.

**Proof** Lemma 27 shows that

$$(1 - 8(\alpha + \frac{1}{c})\gamma)\|y_\theta - \widetilde{x}_\theta\|_2 \le \|y_\theta^* - \widetilde{x}_\theta\|_2 \le (1 + 8(\alpha + \frac{1}{c})\gamma)\|y_\theta - \widetilde{x}_\theta\|_2.$$

Using $\omega$ is non-decreasing and $\omega(\eta \cdot \beta) \le \eta^\gamma \omega(\beta)$, we have

$$(1 - 8(\alpha + \frac{1}{c})\gamma)^\gamma \omega(\|y_\theta - \widetilde{x}_\theta\|_2) \le \omega(\|y_\theta^* - \widetilde{x}_\theta\|_2) \le (1 + 8(\alpha + \frac{1}{c})\gamma)^\gamma \omega(\|y_\theta - \widetilde{x}_\theta\|_2).$$

The result now follows from the assumption $64(\alpha + \frac{1}{c})\gamma^2 \le 1$. ∎

## E.3 Approximate Minimization Regime: when $y_\theta$ is close to $\widetilde{x}_\theta$

In Section E.2, we show that if $\|y_\theta - \widetilde{x}_\theta\|_2$ is large, $\zeta$ approximates $\zeta^*$ up to constant factor. In this section, we handle the other case. We show that if $\|y_\theta - \widetilde{x}_\theta\|_2$ is small, then we can find a $y$ with small function value $g(y)$. First, we show that $\|y_\theta - \widetilde{x}_\theta\|_2$ cannot be too large.

**Lemma 29** *Assume that* $16(\alpha + \frac{1}{c})\gamma \le 1$. *We have*

$$\|y_\theta - \widetilde{x}_\theta\|_2 \le 8c\mu R$$

*for all* $\theta \in [0, 1]$.

**Proof** Case 1: $\omega(\|y_\theta - \widetilde{x}_\theta\|_2) \cdot \|y_\theta - \widetilde{x}_\theta\|_2 \geq c \cdot \delta$. Using this and $16(\alpha + \frac{1}{c})\gamma \leq 1$, Lemma 27 shows that

$$\|y_\theta - \widetilde{x}_\theta\|_2 \leq 2\|y_\theta^* - \widetilde{x}_\theta\|_2. \tag{24}$$

To upper bound $\|y_\theta^* - \widetilde{x}_\theta\|_2$, we use the fact that $y_\theta^*$ is the minimizer of $G_\theta$ and get

$$g(x^*) + W(\|x^* - \widetilde{x}_\theta\|_2) = G_\theta(x^*) \geq G_\theta(y_\theta^*) \geq g(x^*) + W(\|y_\theta^* - \widetilde{x}_\theta\|_2).$$

Since $W$ is increasing, we have $\|y_\theta^* - \widetilde{x}_\theta\|_2 \leq \|x^* - \widetilde{x}_\theta\|_2 \leq \mu R$ where we used Lemma 17. Putting it into (24) gives the result.

Case 2: $\omega(\|y_\theta - \widetilde{x}_\theta\|_2) \cdot \|y_\theta - \widetilde{x}_\theta\|_2 \leq c \cdot \delta$. Since $\delta \leq 8\mu R \cdot \omega(8\mu R)$ and that $\omega$ is increasing, we have that

$$\|y_\theta - \widetilde{x}_\theta\|_2 \leq 8c\mu R.$$

Therefore in both cases we have $\|y_\theta - \widetilde{x}_\theta\|_2 \leq 8c\mu R$ as $c \geq 1$. ∎

Now, we show that small $\|y_\theta - \widetilde{x}_\theta\|_2$ implies small $g(y_\theta)$.

**Lemma 30** *If $\omega(\|y_\theta - \widetilde{x}_\theta\|_2) \cdot \|y_\theta - \widetilde{x}_\theta\|_2 \leq c \cdot \delta$, we have that*

$$g(y_\theta) \leq g(x^*) + \varepsilon.$$

**Proof** By the definition of $y_\theta$ and the assumption, we have

$$\|\nabla g(y_\theta)\|_2 \leq (1+\alpha)\omega(\|y_\theta - \widetilde{x}_\theta\|_2) \cdot \|y_\theta - \widetilde{x}_\theta\|_2 + \delta \leq ((1+\alpha)c + 1)\delta \tag{25}$$

Hence, convexity of $g$ shows that

$$g(y_\theta) - g(x^*) \leq \langle \nabla g(y_\theta), y_\theta - x^* \rangle \leq ((1+\alpha)c + 1)\delta\|y_\theta - x^*\|_2.$$

To bound $\|y_\theta - x^*\|_2$, we note that

$$\|y_\theta - x^*\|_2 \leq \|\widetilde{x}_\theta - x^*\|_2 + \|y_\theta - \widetilde{x}_\theta\|_2 \leq \mu R + 8c\mu R \leq 9c\mu R$$

where we used Lemma 29 and Lemma 17. Hence, convexity of $g$ shows that

$$g(y_\theta) - g(x^*) \leq \langle \nabla g(y_\theta), y_\theta - x^* \rangle \leq ((1+\alpha)c + 1)\delta \cdot 9c\mu R \leq 9c((1+\alpha)c + 1)\varepsilon'.$$

where we used $\delta \leq \frac{\varepsilon'}{\mu \cdot R}$. ∎

## E.4 Bounding Lipschitz constant of $\zeta^*(\theta)$

To derive the stopping criteria $\tau$ (and therefore the iteration complexity), we need to bound the Lipschitz constant of $\zeta^*(\theta)$. We first give an upper bound on $\|\frac{d}{d\theta}(y_\theta^* - \widetilde{x}_\theta)\|$.

**Lemma 31** *We have:*

$$\left\| \frac{d}{d\theta}(y_\theta^* - \widetilde{x}_\theta) \right\| \leq 12\mu\gamma R.$$

**Proof** To compute the derivative of $y_\theta$, we note by optimality condition that

$$\nabla G_\theta(y_\theta^*) = 0.$$

Taking derivatives with respect to $\theta$ on both sides gives

$$\frac{d}{d\theta}\nabla G_\theta(y_\theta^*) + \nabla^2 G_\theta(y_\theta^*) \cdot \frac{d}{d\theta}y_\theta^* = 0.$$

Hence, we have

$$\frac{d}{d\theta}y_\theta^* = -\left(\nabla^2 G_\theta(y_\theta^*)\right)^{-1} \left( (\frac{d}{d\theta}\nabla G_\theta)(y_\theta^*) \right). \tag{26}$$

To bound $\frac{d}{d\theta} y_\theta^*$, we first compute $\frac{d}{d\theta} \nabla G_\theta(y)$ and $\nabla^2 G_\theta(y)$. For $\frac{d}{d\theta} \nabla G_\theta(y)$, we have

$$\frac{d}{d\theta} \nabla G_\theta(y) = \frac{d}{d\theta} \left[ \omega(\|y - \widetilde{x}_\theta\|_2) \cdot (y - \widetilde{x}_\theta) \right]$$

$$= -\omega'(\|y - \widetilde{x}_\theta\|_2) \cdot \frac{(y - \widetilde{x}_\theta)(y - \widetilde{x}_\theta)^\top}{\|y - \widetilde{x}_\theta\|_2} (x^{(2)} - x^{(1)}) - \omega(\|y - \widetilde{x}_\theta\|_2) \cdot (x^{(2)} - x^{(1)}).$$

For $\nabla^2 G_\theta(y)$, Lemma 26 shows that

$$\nabla^2 G_\theta(y) \succeq \omega(\|y - \widetilde{x}_\theta\|_2) \cdot I.$$

Now, (26) shows

$$\|\frac{d}{d\theta} y_\theta^*\| \leq \left[ \frac{\omega'(\|y_\theta^* - \widetilde{x}_\theta\|_2)}{\omega(\|y_\theta - \widetilde{x}_\theta\|_2)} \cdot \left| (y_\theta^* - \widetilde{x}_\theta)^\top (x^{(2)} - x^{(1)}) \right| + \|x^{(2)} - x^{(1)}\|_2 \right]$$

$$\leq \frac{\omega'(\|y_\theta^* - \widetilde{x}_\theta\|_2)}{\omega(\|y_\theta^* - \widetilde{x}_\theta\|_2)} \cdot \|y_\theta^* - \widetilde{x}_\theta\| \cdot \|x^{(2)} - x^{(1)}\| + \|x^{(2)} - x^{(1)}\|_2$$

$$\leq (1 + \gamma) \cdot \|x^{(2)} - x^{(1)}\|_2$$

where we used that $\omega'(s) \leq \gamma \cdot \frac{\omega(s)}{s}$ at the end. Hence, we have

$$\|\frac{d}{d\theta} (y_\theta^* - \widetilde{x}_\theta)\| \leq \|\frac{d}{d\theta} y_\theta^*\| + \|x^{(2)} - x^{(1)}\| \leq (2 + \gamma)\|x^{(2)} - x^{(1)}\|_2.$$

The result follows from $\gamma \geq 1$ and $\|x^{(2)} - x^{(1)}\|_2 \leq 4\mu R$. ∎

We now give a bound on the Lipschitz constant $\zeta^*(\theta)$.

**Lemma 32** *We have*

$$\left| \frac{d}{d\theta} \log \zeta^*(\theta) \right| \leq \frac{2}{1 - \theta} + \frac{1}{\theta} + \frac{12\mu\gamma^2 R}{\|y_\theta^* - \widetilde{x}_\theta\|_2}.$$

**Proof** Note that

$$\frac{d}{d\theta} \log \zeta^*(\theta) = -\frac{2}{1 - \theta} - \frac{1}{\theta} + \frac{\omega'(\|y_\theta^* - \widetilde{x}_\theta\|_2)}{\omega(\|y_\theta^* - \widetilde{x}_\theta\|_2)} \frac{(y_\theta^* - \widetilde{x}_\theta)^\top \frac{d}{d\theta}(y_\theta^* - \widetilde{x}_\theta)}{\|y_\theta^* - \widetilde{x}_\theta\|_2}.$$

Using $\omega'(s) \leq \gamma \cdot \frac{\omega(s)}{s}$, we have

$$\left| \frac{d}{d\theta} \log \zeta^*(\theta) \right| \leq \frac{2}{1 - \theta} + \frac{1}{\theta} + \gamma \frac{\left| (y_\theta^* - \widetilde{x}_\theta)^\top \frac{d}{d\theta}(y_\theta^* - \widetilde{x}_\theta) \right|}{\|y_\theta^* - \widetilde{x}_\theta\|_2^2}$$

$$\leq \frac{2}{1 - \theta} + \frac{1}{\theta} + \gamma \frac{\|\frac{d}{d\theta}(y_\theta^* - \widetilde{x}_\theta)\|_2}{\|y_\theta^* - \widetilde{x}_\theta\|_2}$$

$$\leq \frac{2}{1 - \theta} + \frac{1}{\theta} + \frac{12\mu\gamma^2 R}{\|y_\theta^* - \widetilde{x}_\theta\|_2}$$

from Lemma 31. ∎

Since the Lipschitz constant of $\zeta^*$ depends on the term $\frac{1}{1-\theta}$ and $\frac{1}{\theta}$, we need to show that $\theta$ cannot be too close to 0 and 1. First, we give an upper bound $\theta$.

**Lemma 33 (Upper bound on $\theta$)** *Assume that $16(\alpha + \frac{1}{c})\gamma \leq 1$. For any $\theta \in [0, 1]$ with $\frac{1}{2} \leq \zeta(\theta)$, we have*

$$\theta \leq \max \left( \frac{1}{2}, 1 - \sqrt{\frac{1}{4} \frac{1}{A \cdot \omega(8c\mu R)}} \right)$$

*In particular, we have $u \leq \max\left( \frac{3}{4}, 1 - \frac{1}{2}\sqrt{\frac{1}{4} \frac{1}{A \cdot \omega(8c\mu R)}} \right)$.*

**Proof** Suppose that $\theta \geq \frac{1}{2}$, then we have

$$\frac{1}{2} \leq \zeta(\theta) = \frac{(1-\theta)^2 A}{\theta} \omega(\|y_\theta - \widetilde{x}_\theta\|_2) \leq 2(1-\theta)^2 A \omega(\|y_\theta - \widetilde{x}_\theta\|_2).$$

The bound on $\theta$ now follows from Lemma 29. Since we stop the binary search when $|u - \ell|$ less than $\frac{1}{2} \min \left( \frac{1}{2}, \sqrt{\frac{1}{4} \frac{1}{A \cdot \omega(8c\mu R)}} \right)$, we have the upper bound on $u$. ∎

Next, we give a lower bound on $\theta$.

**Lemma 34 (Lower bound on $\theta$)** *Assume $16(\alpha + \frac{1}{c})\gamma \leq 1$. For any $\theta \in [0,1]$ with $\zeta(\theta) \leq 1$ and $\omega(\|y_\theta - \widetilde{x}_\theta\|_2) \cdot \|y_\theta - \widetilde{x}_\theta\|_2 \geq c \cdot \delta$, we have*

$$\theta \geq \min \left( \frac{1}{2}, \frac{A\delta}{32\mu R} \right).$$

*In particular, we have $\ell \geq \min \left( \frac{1}{4}, \frac{A\delta}{64\mu R} \right)$ or $\omega(\|y_\theta - \widetilde{x}_\theta\|_2) \cdot \|y_\theta - \widetilde{x}_\theta\|_2 \leq c \cdot \delta$.*

**Proof** Suppose that $\theta \leq \frac{1}{2}$, then we have from the assumption

$$
\begin{aligned}
1 \geq \zeta(\theta) &= \frac{(1-\theta)^2 A}{\theta} \omega(\|y_\theta - \widetilde{x}_\theta\|_2). \\
&\geq \frac{1}{4} \cdot \frac{A}{\theta} \omega(\|y_\theta - \widetilde{x}_\theta\|_2) \\
&\geq \frac{1}{4} \cdot \frac{A}{\theta} \frac{c\delta}{\|y_\theta - \widetilde{x}_\theta\|_2} \\
&\geq \frac{1}{4} \cdot \frac{A}{\theta} \frac{c\delta}{8c\mu R}
\end{aligned}
$$

where we used Lemma 29. This gives the lower bound on $\theta$. Since we stop the binary search when $|u - \ell|$ less than $\frac{1}{2} \min \left( \frac{1}{2}, \frac{A\delta}{32\mu R} \right)$, we have the lower bound on $\ell$. ∎

Now, we are ready to show the correctness of Algorithm 4 with the assumed $\tau$.

**Theorem 35 (Correctness of Algorithm)** *Assume $64(\alpha + \frac{1}{c})\gamma^2 \leq 1$. Algorithm 4 outputs either $y$ such that*

$$g(y) \leq g^* + \varepsilon$$

*or $y = y_\theta$ such that*

$$\frac{1}{2} \leq \zeta(\theta) \leq 1$$

*with*

$$\|\nabla g(y_\theta) + \omega(\|y_\theta - \widetilde{x}_\theta\|_2) \cdot (y_\theta - \widetilde{x}_\theta)\| \leq \alpha \cdot \omega(\|y_\theta - \widetilde{x}_\theta\|_2) \cdot \|y_\theta - \widetilde{x}_\theta\|_2 + \delta$$

*where $\delta \leq \frac{1}{c}\omega(\|y_\theta - \widetilde{x}_\theta\|_2) \cdot \|y_\theta - \widetilde{x}_\theta\|_2$.*

**Proof** For the case $\omega(\|y_\ell - \widetilde{x}_\ell\|_2) \cdot \|y_\ell - \widetilde{x}_\ell\|_2 \leq c \cdot \delta$ and $\omega(\|y_u - \widetilde{x}_u\|_2) \cdot \|y_u - \widetilde{x}_u\|_2 \leq c \cdot \delta$, Lemma 30 shows that $g(y) \leq g^* + \varepsilon$.

Otherwise, Lemma 33 and Lemma 34 show that

$$\ell \geq \min \left\{ \frac{1}{4}, \frac{A\delta}{64\mu R} \right\} \tag{27}$$

and

$$u \leq \max \left\{ \frac{3}{4}, 1 - \frac{1}{2} \sqrt{\frac{1}{4} \frac{1}{A \cdot \omega(8c\mu R)}} \right\}. \tag{28}$$

850 Therefore, together with Lemma 32 we have

$$\left| \frac{d}{d\theta} \log \zeta^*(\theta) \right| \leq \frac{2}{1-\theta} + \frac{1}{\theta} + \frac{12\mu\gamma^2 R}{\|y_\theta^* - \widetilde{x}_\theta\|_2}$$

$$\leq 12 + 4\sqrt{4A \cdot \omega(8c\mu R)} + \frac{64\mu R}{A\delta} + \frac{12\mu\gamma^2 R}{\|y_\theta^* - \widetilde{x}_\theta\|_2} \tag{29}$$

851 for all $\ell \leq \theta \leq u$. To bound the term $\|y_\theta^* - \widetilde{x}_\theta\|_2$, note from Lemma 28 we have

$$\frac{8}{7}\|y_u^* - \widetilde{x}_u\|_2 \geq \|y_u - \widetilde{x}_u\|_2 \geq \frac{c\delta}{\omega(\|y_u - \widetilde{x}_u\|_2)}. \tag{30}$$

852 Using $\frac{3}{4} \geq \zeta(u)$ (due to binary search), we have

$$\frac{3}{4} \geq \zeta(u) = \frac{(1-u)^2 A}{u}\omega(\|y_u - \widetilde{x}_u\|_2) \geq (1-u)^2 A\omega(\|y_u - \widetilde{x}_u\|_2).$$

853 Putting it into (30) gives

$$\|y_u^* - \widetilde{x}_u\|_2 \geq \frac{28c\delta(1-u)^2 A}{24} \geq \frac{7c\delta A}{6}\frac{1}{16A \cdot \omega(8c\mu R)} \geq \frac{c\delta}{15 \cdot \omega(8c\mu R)}$$

854 where we used (28) for the last inequality. Lemma 31 shows that

$$\left\| \frac{d}{d\theta}(y_\theta^* - \widetilde{x}_\theta) \right\| \leq 12\mu\gamma R.$$

855 Since we have from the stopping criteria $\tau = |u - \ell| \leq \frac{c\delta}{360\mu\gamma R \cdot \omega(8c\mu R)}$, for all $\ell \leq \theta \leq u$, this gives

$$\|y_\theta^* - \widetilde{x}_\theta\|_2 \geq \|y_u^* - \widetilde{x}_u\|_2 - 12\mu\gamma R \cdot \tau \geq \frac{c\delta}{30\omega(8c\mu R)}.$$

856 Put together with (29) we have

$$\left| \frac{d}{d\theta} \log \zeta^*(\theta) \right| \leq 12 + 8\sqrt{A \cdot \omega(8c\mu R)} + \frac{64\mu R}{A\delta} + \frac{360\mu\gamma^2 R\omega(8c\mu R)}{c\delta}$$

$$\leq 20 + 20A \cdot \omega(8c\mu R) + \frac{64\mu R}{A\delta} + \frac{6\mu R\omega(8c\mu R)}{\delta}$$

$$\leq 20\left( 1 + A \cdot \omega(8c\mu R) + \frac{4\mu R}{A\delta} + \frac{\mu R}{\delta} \cdot \omega(8c\mu R) \right)$$

857 where we used $64(\alpha + \frac{1}{c})\gamma^2 \leq 1$ and $\alpha \leq 1$. Due to the choice of $\tau \leq$

858 $\frac{1}{200\left(1 + A \cdot \omega(8c\mu R) + \frac{4\mu R}{A\delta} + \frac{\mu R}{\delta} \cdot \omega(8c\mu R)\right)}$, this shows that $\zeta^*(\ell) \leq e^{\frac{1}{10}}\zeta^*(u)$. Now, using Lemma 28,

859 we have

$$\zeta(\ell) \leq \frac{8}{7}\zeta^*(\ell) \leq \frac{8}{7}e^{\frac{1}{10}}\zeta^*(u) \leq \frac{8}{7}e^{\frac{1}{10}}\frac{5}{4} \cdot \zeta(u) \leq \frac{8}{7}e^{\frac{1}{10}}\frac{5}{4}\frac{3}{4} \leq 1.$$

860 Moreover, by the definition of binary search, we know $\zeta(\ell) \geq \frac{3}{4}$. This completes the proof that we
861 have found a point satisfying $\frac{1}{2} \leq \zeta(\theta) \leq 1$. ∎

862

863 ## E.5  Bounding the number of steps

864 To bound the number of steps, we need to have a lower and upper bound on $A$. We note that when
865 we apply the line search procedure, we have $A = A_k$ at iteration $k$. Furthermore, we assume $k \geq 1$
866 because no line search is needed for $k = 0$. Under the assumption, we have $\frac{1}{2\omega(2\mu R)} \leq A \leq \frac{R^2}{\varepsilon}$.
867 Below we give the proof of the main theorem for the line search implementation.

868 **Proof** [Proof of Theorem 19] Recall from the algorithm description, we set

$$\frac{1}{\tau} \le 4 + 2\sqrt{4A \cdot \omega(8c\mu R)} + \frac{64\mu R}{A\delta} + \frac{360\mu\gamma R \cdot \omega(8c\mu R)}{c\delta}$$
$$+ 200\left(1 + A \cdot \omega(8c\mu R) + \frac{4\mu R}{A\delta} + \frac{\mu R}{\delta} \cdot \omega(8c\mu R)\right)$$
$$\le 300\left(1 + A \cdot \omega(8c\mu R) + \frac{4\mu R}{A\delta} + \frac{\mu R}{\delta} \cdot \omega(8c\mu R)\right)$$

869 where we used $16(\alpha + \frac{1}{c})\gamma \le 1$. Now using $\frac{1}{2\omega(2\mu R)} \le A \le \frac{R^2}{\varepsilon}$ from the assumption we get

$$\frac{1}{\tau} \le 300\left(1 + \left(\frac{R^2}{\varepsilon} + \frac{9\mu R}{\delta}\right) \cdot \omega(8c\mu R)\right)$$
$$\le 40\left[\frac{160\mu Rc}{\delta} + \frac{9R^2}{\varepsilon}\right] \cdot \omega(8c\mu R)$$

870 where we used $\delta \le 8\mu R \cdot \omega(8\mu R)$ at the end. Putting together with Theorem 35 yields the result. ∎

871