[Reviews · NeurIPS 2019]

Reviewer 1



Quality: The submission is purely theoretical supported by solid analysis. I do not check the proofs in the appendix, but the claims sound correct to me. However, the constants in the almost all theorems seem to be too complicated to be the inherent constants for the problem. Originality: The lower bound for parallel non-smooth optimization has not been very well-studied and the submission presents a fairly thorough answer to that for both upper and lower bound. Clarity: Although the problem setting is clear, it is hard to understand the proof techniques in the lower bound part. More explanations perhaps with a figure for the wall function will be helpful. Significance: Obtaining the lower bound and constructing the random wall function are non-trivial in theory. Although the proposed algorithm seems to be impractical, the results provide a guideline on the lower bound for other algorithms fallen in the regime. Updates after rebuttal: In Theorem 7, usually \eps<1, then it requires \delta<10^{-20}, which suggests the algorithm needs an exact gradient oracle.

Reviewer 2



The work is very original and provides novel approaches for both the lower bound problem, where the new idea of a wall function is introduced, and for the upper bound side, where the authors deploy a new framework for the design of accelerated algorithms. Related work is adequately cited and the key ideas in previous works are properly summarized and explained, which makes it easier to follow the novel technical arguments. The results are significant. The authors also provide a number of open problems and conjectures to build on.

Reviewer 3



# Setting Suppose we want to minimize a non-smooth convex function over d dimensions. At each round, we can query poly(d) function values in parallel (i.e. highly parallel). The goal is to reduce the number of total interactive rounds. # Significance (see also the contributions section) 1. The lower bound construction which shows that that subgradient descent is optimal up to O(d^{1/2}) is a refinement of (and the result an improvement upon) a decades-old lower bound by Nemirovski which shows up to T = O(d^{1/3}). This closes the question of up to what T is subgradient descent optimal (i.e. parallelization does not help). 2. The upper bound result of d^{1/2}/\eps^{2/3} interpolates between subgradient descent (with the complexity of 1/\eps^{2}) and interior-point methods with the complexity of d\log(\eps). This improves upon the previous best upper bound of d^{1/4}/\eps, and the authors conjecture to be optimal. # Things to improve 1. [Clarity] The lower bound construction is explained very well and is easy to read. A formal statement of the theorem in the main paper would add further clarity. On the other hand, the writing of the upper bound seems a little hurried---while oracles (ll 238--249) are defined their significance is never discussed and neither is their implementation, (ll 208--233) discusses in detail a method which does not work, the actual algorithm which does work is never explicitly stated. In general, section 3 discusses many pieces which are used for the final algorithm but does not talk about the actual method. The authors should consider reorganizing their writing so that the high-level details of the method are apparent in the main paper and the implementation details of the pieces are moved to the appendix. 2. It is unclear from the discussion if for \eps smaller than (1/d) if center of gravity method is optimal. Does parallelization not help in this case either (this seems to be implied by the discussion in Sec 1.2)? 3. Line 63 "in the range [d^{-1}, d^{-1/2}]" -> "in the range [d^{-1}, d^{-1/4}]" 4. For \eps in the range [d^{-1}, d^{-1/4}], authors conjecture that d^{1/3}/\eps^{2/3} is optimal. A discussion of their intuitions (perhaps in the appendix) on why this is true would be helpful. 5. It is unclear why the function \Chi is used in algorithm 1 (especially since it increases the bias of the estimator). The proof in the appendix does not make it clear what properties of \Chi are necessary (e.g. why would a gaussian kernel not work...). ==== Update after reading rebuttal and discussion === I thank the authors for their reply. I have one additional concern/recommendation for the writing: lemma 8 only establishes only 1-order smoothness of the function and not p-th order as is required for the acceleration framework. Adding this would aid understanding. I also realized I made an error of assuming that the oracle used is a zero-order one instead of a first order one. However using d parallel function queries one can compute the gradient of the smoothed function to arbitrary accuracy and so leaves their results (and my review) unchanged.

[Author Response · NeurIPS 2019]

**Reviewer #1:**

We did our best to explain the lower bound construction in the most pedagogically principled way. In fact we also tried to create a 2-dimensional picture to illustrate our construction, however decided against it as the phenomenon at play seems to be inherently high-dimensional and low-dimensional pictures are misleading (in particular it is not possible to represent the "valley" to the optimum in low dimensions). If you have other suggestions to clarify the construction we would be happy to hear them!

We also would like to point out that, while you are certainly correct that there are only few papers on lower bounds for parallel non-smooth optimization, it might be misguided to characterize the problem as "not very well-studied". There are thousands of papers on variants of this topic, and arguably the reason for the paper shortage on the most canonical variant (the one studied in this paper) is that progress was hard to come by after Nemirovski's seminal contributions from the 90s. We believe that our new ideas will be of value to both the machine learning community and to the broader optimization community, despite the fact that the current algorithm is not practical.

**Reviewer #2:**

Thank you for the kind words!

**Reviewer #3:**

Thank you for your positive feedback! Here is a detailed answer to your questions:

1. Thank you for the suggestion, we will consider re-organizing the material (it is a challenge to make this fit in 8 pages!).

2. This is in fact a great open problem! As far as we know, for the tiny epsilon regime (say $1/\text{poly}(d)$), the best lower bound used to be Nemirovski's $d^{1/3}$, which is now $d^{1/2}$ thanks to the wall function argument of our paper. We will clarify this point in the paper.

3. Agreed, thanks!

4. The intuition is actually quite simple: line 139 on page 4 shows that the lower bound construction would imply the $d^{1/3}/\epsilon^{2/3}$ if one could take delta (the radius outside of which the "walling" starts) to be epsilon. This is technically difficult because for such some delta, our proposed wall function will "leak information" between dimensions. However it seems believable that there could be a more complicated wall function that could work at such tiny walling radii. We will add a comment along those lines in the paper.

5. You are correct that a Gaussian kernel in lieu of $\chi$ could possibly work, however as far as we can see this would make the proof longer. Our current proof crucially relies on the fact that the tail of $\chi$ is zero, which wouldn't be true with a Gaussian kernel.

[Meta-Review · NeurIPS 2019]

The work is more an exploration of an oracle model for optimization rather than designing new algorithms. While the results are not practical, the work largely answers the question of what is possible with the chosen oracle model. The paper in the end provides a 'target' complexity so that other researchers may design more practical methods which match the theoretical bounds of this work. The reviewers unanimously agree that the work should be accepted. The authors' rebuttal also helped clarify some concerns and ultimately increased the overall score of the work.